# Regulation of PCNA cycling on replicating DNA by RFC and RFC-like complexes

Mi-Sun Kang [ID] [1,4], Eunjin Ryu[1,2,4], Seung-Won Lee[2,4], Jieun Park[1], Na Young Ha[1], Jae Sun Ra[1], Yeong Jae Kim[1,2], Jinwoo Kim[1,2], Mohamed Abdel-Rahman[3], Su Hyung Park[1], Kyoo-young Lee[1], Hajin Kim[1,2], Sukhyun Kang[1] & Kyungjae Myung[1,2]

Replication-Factor-C (RFC) and RFC-like complexes (RLCs) mediate chromatin engagement of the proliferating cell nuclear antigen (PCNA). It remains controversial how RFC and RLCs cooperate to regulate PCNA loading and unloading. Here, we show the distinct PCNA loading or unloading activity of each clamp loader. ATAD5-RLC possesses the potent PCNA unloading activity. ATPase motif and collar domain of ATAD5 are crucial for the unloading activity. DNA structures did not affect PCNA unloading activity of ATAD5-RLC. ATAD5-RLC could unload ubiquitinated PCNA. Through single molecule measurements, we reveal that ATAD5-RLC unloaded PCNA through one intermediate state before ATP hydrolysis. RFC loaded PCNA through two intermediate states on DNA, separated by ATP hydrolysis. Replication proteins such as Fen1 could inhibit the PCNA unloading activity of Elg1-RLC, a yeast homolog of ATAD5-RLC in vitro. Our findings provide molecular insights into how PCNA is released from chromatin to finalize DNA replication/repair.

[1] Center for Genomic Integrity, Institute for Basic Science, Ulsan 44919, Republic of Korea. [2] School of Life Sciences, Ulsan National Institute of Science and Technology, Ulsan 44919, Republic of Korea. [3] Department of Ophthalmology, Ohio State University Comprehensive Cancer Center, Columbus, OH 43210, USA. [4]These authors contributed equally: Mi-Sun Kang, Eunjin Ryu, Seung-Won Lee. Correspondence and requests for materials should be addressed to H.K. (email: hajinkim@unist.ac.kr) or to S.K. (email: kangsh@ibs.re.kr)

DNA replication is an essential process for transfer of genetic information from a parent to daughter cells and should be tightly regulated[1,2]. Imprecise DNA replication could result in genomic instability. Regulation of DNA replication is mainly achieved by timely assembly and disassembly of replication machineries on the chromatin[3]. Therefore, the completion of DNA synthesis needs to be tightly correlated with the disassembly of replication machineries from chromosomal DNA.

Loading of proliferating cell nuclear antigen (PCNA) onto the chromatin is a crucial step for the initiation of DNA synthesis[4]. PCNA, a homo-trimeric and ring-structured molecule, encircles DNA[5] and functions as a clamp for DNA polymerases[6]. PCNA functions as a molecular platform for DNA replication, repair, and chromatin assembly[7–10]. Thus, precise control of PCNA loading and unloading is critical for efficient DNA replication/repair.

Given its ring structure, PCNA needs to be opened and closed to encircle the DNA[11]. Replication factor C (RFC), a pentameric AAA+ ATPase complex, which consists of RFC1–5, loads PCNA to the primer-template junctions. Based on various experimental results, a model for RFC-mediated PCNA loading on the primed-template DNA has been suggested[12–14]. In this model, a spiral-shaped RFC complex assembles with PCNA in the presence of ATP, opening a gap in the trimeric ring. Upon DNA binding, the opened PCNA-ring twists like a spring washer and closes after ATP hydrolysis, followed by RFC dissociation from DNA-loaded PCNA. It was reported that the opening of PCNA by RFC upon ATP binding and its closure upon ATP hydrolysis are the rate-limiting steps during yeast PCNA loading[15]. The loading dynamics of human PCNA have not yet been clearly assessed by single molecule measurements.

After chromosomal duplication, PCNA needs to be released from the chromatin to prevent inappropriate recruitment of replication enzymes[3,16,17]. RFC-like complexes (RLCs) are suggested to regulate chromatin-association of DNA clamps. RLCs share four small subunits—RFC2, 3, 4, and 5—with RFC. The largest subunit of RFC, RFC1, is replaced by different proteins in RLCs. Mammalian cells have three RLCs: ATAD5-RLC, CTF18-RLC, and RAD17-RLC. However, it has been controversial which complex is responsible for the release of DNA-loaded PCNA. Previously, it was reported that human RFC could unload PCNA[18–21]. Another report argued that yeast Ctf18-RLC might have PCNA unloading activity[22,23]. However, those in vitro experiments have not been supported by cell-based experiments. RAD17-RLC loads DNA-damage clamps[24]. PCNA unloading could be mediated by ATAD5-RLC in human cells[3]. Depletion of human ATAD5 led to PCNA accumulation on chromatin. In budding yeast, deletion of elg1, a homolog of ATAD5, also resulted in an increase of chromatin-bound PCNA[17,25,26]. Replication proteins that interact with PCNA abnormally accumulated on the chromatin in ATAD5-depleted human cells, showing the importance of PCNA unloading[3]. Furthermore, elg1 or ATAD5 depletion caused genomic instability[27,28]. These results imply the functional importance of ATAD5-RLC, but it has not been proven whether ATAD5-RLC is a biochemically defined PCNA unloader. Furthermore, there have been no studies on the mechanism of PCNA unloading, which, due to the irreversible step of ATP hydrolysis, cannot simply be the reversal of the PCNA loading reaction.

PCNA unloading should be tightly regulated to prevent premature termination of DNA replication. It has been reported that the ligation of Okazaki fragments is required for PCNA removal from replicated chromatin in yeast[29]. However, the molecular basis for timely PCNA unloading remains to be elucidated.

In addition to PCNA unloading, ATAD5 participates in the de-ubiquitination of ubiquitinated PCNA (Ub-PCNA)[30]. PCNA is mono- or poly-ubiquitinated when the replication fork is stalled by DNA lesions[31–35]. After bypass, chromatin-bound Ub-PCNA needs to be removed from DNA to resume normal DNA replication. However, it is still unknown whether Ub-PCNA can be unloaded from DNA in an ubiquitinated form by certain RLCs.

Here we discover that human ATAD5-RLC has the most potent PCNA unloading activity among clamp-loader complexes. PCNA unloading by ATAD5-RLC shows unanticipated characteristics. DNA structures that mimic ongoing DNA replication, do not affect the PCNA unloading activity of ATAD5-RLC. On the other hand, we find that yeast replication enzymes such as Fen1 inhibits PCNA unloading by Elg1-RLC in vitro. Furthermore, ATAD5-RLC efficiently unloads mono- and poly-ubiquitinated PCNA. Moreover, we employ single molecule FRET (smFRET) techniques and revealed distinct intermediate states during PCNA loading and unloading. This study gives critical insights to how replication termination is regulated by the controlled dissociation of PCNA from replicated chromatin.

## Results

**ATAD5-RLC is a potent PCNA unloader.** To assess PCNA unloading activities of RFC and RLCs, we set up in vitro PCNA loading and unloading reactions with RFC and RLCs that are purified using Baculovirus system (Supplementary Fig. 1a). In the case of RFC, we used N-terminal-deleted RFC1-RFC (RFC1 (ΔN554)-RFC) that has robust PCNA loading activity (Supplementary Fig. 1b). First, we mapped a minimal region of ATAD5 sufficient for PCNA unloading. ATPase domain of ATAD5 is located at its C-terminal half (Fig. 1a). ATAD5 contains a long N-terminal region for Ub-PCNA de-ubiquitination[30]. Functional contribution of the N-terminal domain for PCNA unloading has not been known. We prepared a series of ATAD5 deletion mutants to identify a functional domain that is responsible for PCNA unloading (Fig. 1a). A putative nuclear localization signal (NLS) sequence is located at the N-terminus of ATAD5 (amino-acid residues 1–32). The NLS sequence was fused to each N-terminal deletion mutant to prevent mis-localization. Each ATAD5 deletion mutant was expressed in ATAD5-depleted cells, and the amount of chromatin-bound PCNA was monitored (Fig. 1b, c). As previously reported, depletion of ATAD5 in 293T cells increased the amount of chromatin-bound PCNA. Add-back of wild-type ATAD5 reduced PCNA levels on chromatin. Deletion of N-terminus of ATAD5 up to 692 amino acids (ΔN692) did not affect PCNA unloading activity. These results suggest that the PCNA unloading activity of ATAD5 is conferred by its C-terminal domain and can be functionally separated from its de-ubiquitination function.

Based on the above results, we purified the N-terminal-deleted ATAD5 (ATAD5 (ΔN692)) in complex with RFC2–5 using the Baculovirus expression system (Fig. 1d). Purified ATAD5 (ΔN692)-RLC had a stoichiometric amount of each subunit. To examine PCNA unloading by each clamp-loader complex, an in vitro PCNA unloading assay was established (Fig. 1e). We prepared magnetic bead-attached 130-mer DNA that contains 10 nucleotides gap. Free end of DNA was blocked by TALE protein to prevent PCNA from sliding off. PCNA was loaded onto the substrate DNA using purified RFC. After wash, clamp-loader complexes were added to the DNA-loaded PCNA. ATAD5-RLC removed PCNA from the substrate DNA in a concentration-dependent manner (Fig. 1f). Approximately 65 fmol of PCNA bound to 0.5 pmol of substrate DNA (Supplementary Fig. 1b), and 25 fmol (0.6 nM in 40 μL reaction) of ATAD5 (ΔN692)-RLC was sufficient to remove most PCNA from DNA. However, RFC, CTF18-RLC or RAD17-RLC did not reduce PCNA on DNA. CTF18-RLC loaded PCNA and RAD17-RLC loaded the 9-1-1

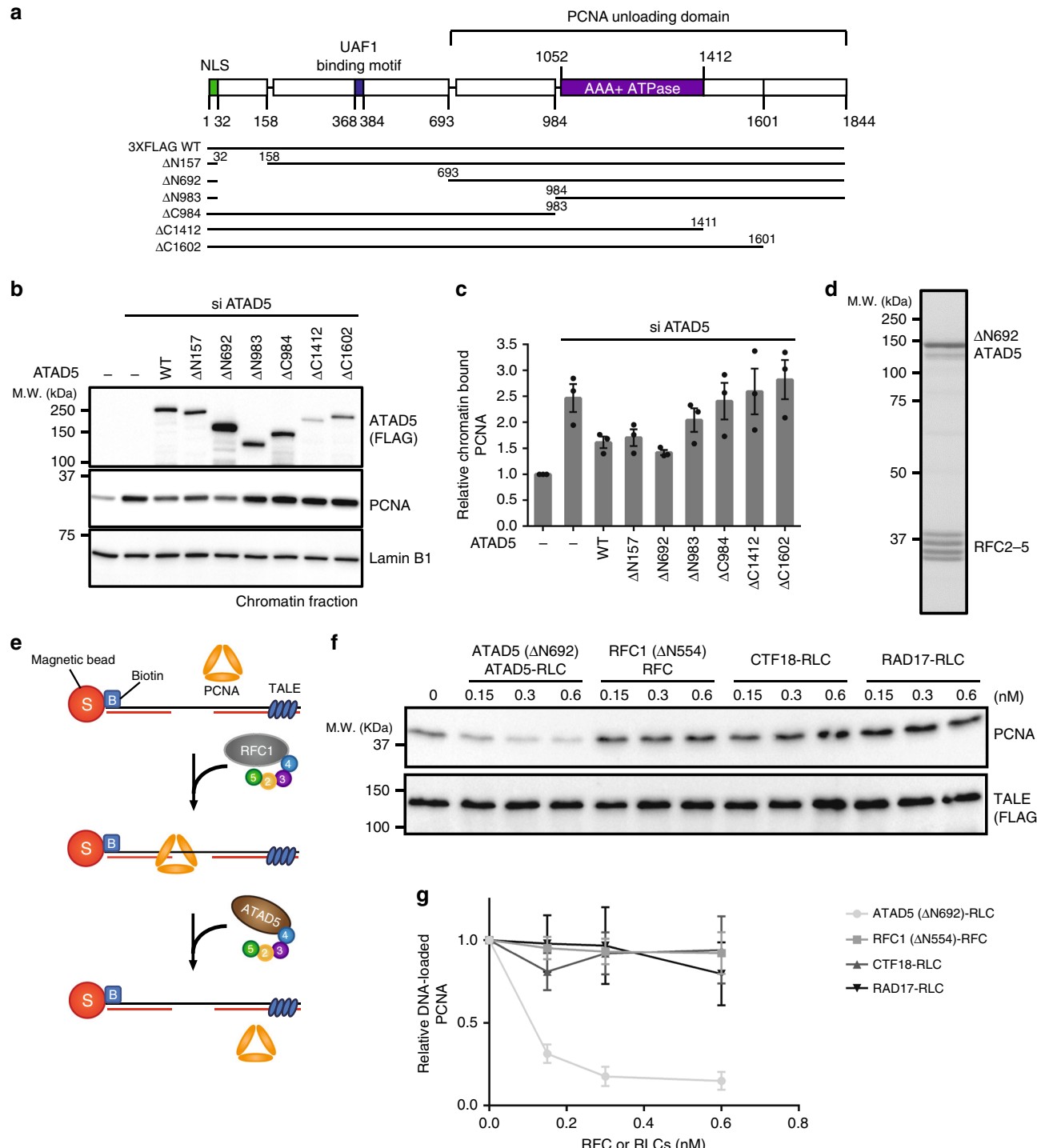

complex to the DNA, as previously reported (Supplementary Fig. 1c and d, respectively)[23,24]. The PCNA unloading activity of ATAD5 (ΔN692)-RLC was also observed with longer DNA substrate that contains nine primer-template junctions (Supplementary Fig. 1e). We prepared RFC containing full-length RFC1 using the combination of yeast and bacterial expression system (Supplementary Fig. 1a). Full-length RFC loaded PCNA (Supplementary Fig. 1c), but did not reduced DNA-loaded PCNA (Supplementary Fig. 1f). ATAD5 (ΔN692)-RLC did not unload DNA-bound 9-1-1 complex (Supplementary Fig. 1g). These results show that ATAD5-RLC is a potent PCNA unloading complex.

PCNA unloading should not occur during DNA synthesis to prevent futile PCNA loading and unloading cycle. Because replicating DNA intermediates are structurally different from fully replicated double-stranded DNA, DNA structure might affect the PCNA unloading activity. Therefore, we tested PCNA unloading from DNA substrates mimicking steps of lagging strand synthesis (Supplementary Fig. 1h–j). First, we prepared two short 80-mer DNA substrates, which contained different size of gap—1 or 11 nucleotides, respectively (Supplementary Fig. 1h). ATAD5 (ΔN692)-RLC unloaded PCNA from those two substrates with similar efficiency (Supplementary Fig. 1i). To examine whether fully duplexed DNA enhances PCNA unloading

**Fig. 1** ATAD5-RLC is a robust PCNA unloader. **a** Deletion mutants of ATAD5 examined for PCNA unloading activity. Top diagram shows the locations of the reported motifs in human ATAD5. Lines between boxes represent predicted coiled-coil structures. Boundaries of motifs are denoted as amino-acid numbers. N-terminal deletion mutants of ATAD5 were fused to its putative NLS sequence located at its N-terminus (residues 1–32 of ATAD5). **b** The PCNA unloading activity is conferred by the C-terminal domain of ATAD5. The indicated ATAD5 deletion mutants were transiently expressed in ATAD5-depleted 293T cells. Chromatin-bound PCNA was analyzed by immunoblotting after chromatin fractionation. ATAD5 (ΔN692) exhibited PCNA unloading activity. **c** Quantification of chromatin-bound PCNA in (**b**). Values indicate relative chromatin-bound PCNA amount compared with the control chromatin ($n = 3$). Error bars indicate standard deviations in all quantification results. **d** Purification of ATAD5-RLC. Baculovirus encoding HIS-ATAD5 (ΔN692)-3XFLAG were co-infected to insect cells with a virus encoding RFC2–5. Pentameric ATAD5-RLC was purified from cell extracts through sequential application onto HIS, FLAG, and Heparin affinity columns. Coomassie-stained purified ATAD5-RLC has a stoichiometric amount of each subunit. **e** Schematic diagram of PCNA unloading assay. DNA substrates were prepared by annealing oligonucleotides to the 5′-biotinylated DNA. Substrate DNA was attached to the streptavidin-coated magnetic beads. PCNA was loaded to the substrate DNA by purified RFC. After washing, ATAD5-RLC was treated to the DNA-loaded PCNA. PCNA that remained on DNA was analyzed by immunoblotting. Error bars indicate standard deviations in all PCNA loading/unloading results. **f** ATAD5-RLC unloads PCNA. Indicated amounts of purified ATAD5-RLC, RFC, CTF18-RLC, and RAD17-RLC were treated to the DNA-loaded PCNA. 10-nucleotide-gap DNA (130-mer) was used for this assay. Unloading reaction contained 1 mM ATP. **g** Quantification of (**f**). Relative DNA-loaded PCNA amounts are indicated compared with the control reaction. Error bars indicate standard deviation ($n = 3$). See also Supplementary Fig. 1

by ATAD5 (ΔN692)-RLC, we prepared DNA substrate containing single nick (Supplement Fig. 1j). After PCNA loading, T4 DNA ligase was added to the reaction to ligate DNA fragments. Addition of T4 DNA ligase did not affect PCNA unloading activity of ATAD5 (ΔN692)-RLC. These results imply that DNA structure does not affect PCNA unloading activity of ATAD5-RLC.

Because previous studies reported PCNA unloading activity of RFC[18–21], we carefully compared PCNA unloading activity of ATAD5 (ΔN692)-RLC and RFC1 (ΔN554)-RFC purified from the Baculovirus expression system (Supplementary Fig. 1k–n). First, we reduced ATP concentration in PCNA unloading reaction, since it was reported that high ATP concentration could inhibit PCNA unloading activity of RFC[20]. However, the amount of DNA-loaded PCNA was not reduced by RFC at 0.5 mM or 1 mM ATP (Supplementary Fig. 1k). PCNA unloading activity of ATAD5-RLC was not affected by ATP concentration. Increase of RFC in unloading reaction reduced the amount of DNA-loaded PCNA (Supplementary Fig. 1l). However, PCNA unloading activity of RFC was significantly lower than that of ATAD5-RLC (compare Supplementary Fig. 1k and Supplementary Fig. 1l). RFC unloaded PCNA more efficiently from the nicked DNA compared with 10-nucleotide-gap DNA (Supplementary Fig. 1m). Even with nicked DNA, PCNA unloading activity of RFC was significantly lower than that of ATAD5 (Supplementary Fig. 1n). These results confirm that ATAD5-RLC is the most potent PCNA unloader among clamp-loader complexes.

**Proper RLC assembly is crucial for PCNA unloading**. The C-terminus of RFC1 functions as a RFC2–5 binding motif (RBM) that gathers RFC subunits together[12]. Consistently, C-terminal-deleted RFC1 failed to interact with RFC2–5 (Supplementary Fig. 2a). The C-terminus of ATAD5 is also important for RLC formation with RFC2–5 (Fig. 2a–c). The 243 residues of the C-terminal (residues 1602–1844) of ATAD5 expressed in 293T cells were able to bind to RFC subunits (Fig. 2b). Deletion of the first half of this region (residues 1602–1719) abolished the interaction with RFC2–5. Deletion of the second half (residues 1720–1844) also resulted in failure to interact with RFC subunits (Fig. 2c). Therefore, the C-terminus ATAD5 (residues 1602–1844) functions as an RBM.

There are several well-conserved regions in the RBM of ATAD5 (Supplementary Fig. 2b). We mutated four conserved motifs of five-amino-acid stretches to alanine or glycine. Among them, $E^{1713}$ALSF$^{1717}$ to AAGGG mutant (CM1) was severely defective in binding to RFC2–5 (Fig. 2d). This result implies that $E^{1713}$ALSF$^{1717}$ of ATAD5 either directly interacts with RFC2–5

or is important for the formation of the structure that is crucial for RFC2–5 binding. Interestingly, RLC-formation-defective mutants also showed significantly reduced PCNA unloading activity (ΔC1412 and ΔC1602 in Fig. 1b, c, ATAD5 (693–1719) in Fig. 2e and ATAD5 (CM1) in Fig. 2f). Although other CM mutants (CM2-CM4) were not defective in RFC2–5 binding (Fig. 2d), those mutants showed different levels of PCNA unloading defects (Fig. 2f). These results suggest that those mutations in RBM might induce conformational changes of ATAD5-RLC, which compromise its catalytic activity. Furthermore, when the RBM of ATAD5 (residues 1601–1844) was swapped with that of RFC1 (residues 835–1147; ATAD5 (ΔN692 CR)), PCNA unloading activity was significantly reduced (Supplementary Fig. 2c–e), although the swap mutant was still able to form a complex with RFC2–5 (Supplementary Fig. 2d). This result suggests that the RBM of ATAD5 is important for forming the active conformation of ATAD5-RLC to unload PCNA.

Next, we purified an RLC-formation-defective mutant (693–1719) of ATAD5 using the Baculovirus expression system (Supplementary Fig. 2f). Because ATAD5 (693–1719) was less stable in insect cells, we purified ATAD5 (693–1719) and ATAD5 (ΔN692)-RLC using the combination of yeast and bacterial expression system (Supplementary Fig. 2g). ATAD5 (693–1719) bound to a lesser amount of RFC2–5 compared with ATAD5 (ΔN692) and was defective in PCNA unloading in vitro (Fig. 2g (purified from Baculovirus system) and Supplementary Fig. 2h (purified from yeast-bacterial system)). Next, we purified ATAD5 (ΔN692) alone and full-length ATAD5-RLC using the yeast-bacterial system (Supplementary Fig. 2i) and compared their PCNA unloading activity with ATAD5 (ΔN692)-RLC purified from the same system (Supplementary Fig. 2j). ATAD5 (ΔN692) alone did not show PCNA unloading activity. PCNA unloading activity of full-length ATAD5-RLC and ATAD5 (ΔN692)-RLC were similar. Therefore, N-terminal domain of ATAD5 is not essential for PCNA unloading and PCNA unloading requires ATAD5-RLC, not ATAD5 alone.

**The ATPase interface of ATAD5 regulates PCNA unloading**. AAA+ ATPase family proteins have conserved Walker A and Walker B motifs in ATP binding pocket[36]. The lysine residue of the Walker A motif is critical for ATP binding, and the glutamate or aspartate residue of the Walker B motif is known to be important for ATP binding and/or hydrolysis. In ATAD5, the essential lysine residue ($K^{1138}$) of Walker A ($G^{1132}$PTGVG$\underline{K}T^{1139}$) is well conserved (Fig. 3a). Although less conserved, we could find a putative Walker B motif within residues 1301–1308 of ATAD5 ($L^{1301}$ILFEEVD$^{1308}$).

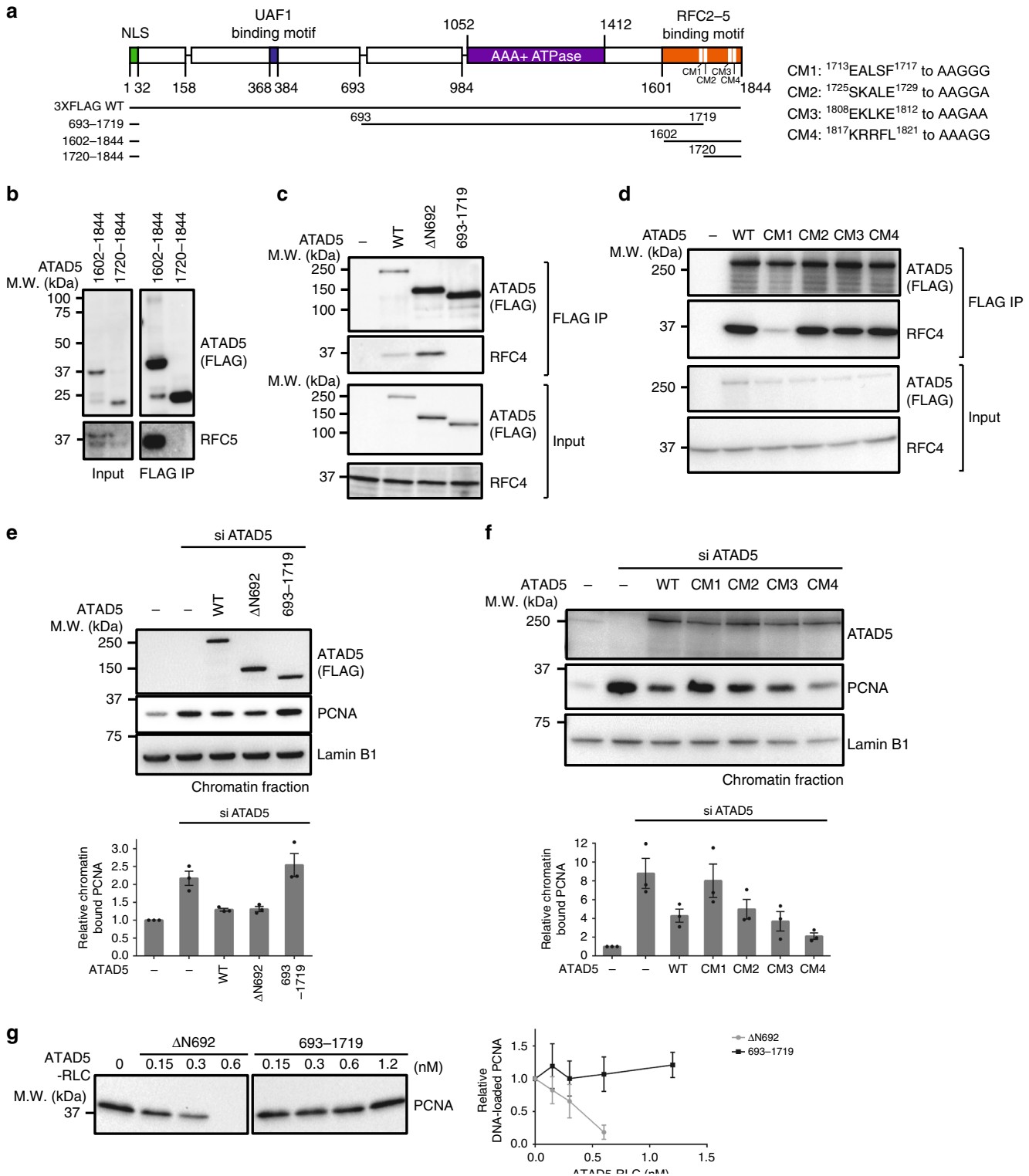

We prepared ATAD5 variants that have mutations in essential residues of the Walker A or Walker B motif, denoted as KA and NQA mutants, respectively (Fig. 3a and Supplementary Fig. 3a, b). There is a conserved polypeptide stretch between the Walker A and Walker B motifs (Supplementary Fig. 3a). We recently identified a germline mutation E1173K (denoted as EK mutant) in this region in a proband and his mother both with uveal (intraocular) melanoma. We examined the PCNA unloading

activity of these variants in ATAD5-depleted 293T cells (Fig. 3b). We used the short form of ATAD5 where residues 33–399 were deleted (ATAD5 (ΔN400)). Interestingly, KA, NQA, and EK mutants were all defective in PCNA unloading. ATPase motifs of AAA+ ATPase family proteins were located at the interface between two neighboring subunits. Therefore, ATPase-motif mutations often affect the integrity of the complex. Both KA and NQA mutants showed reduced binding to RFC3, 4, and 5, but

**Fig. 2** Active ATAD5-RLC formation requires functional RFC2–5 binding motif. **a** Constructs to identify the RFC2–5 binding motif (RBM). The positions of the deletions and point mutations (CM1–4) are indicated. Four conserved five-amino-acid stretches in RBM were mutated and denoted as CM1–4 (see Supplementary Fig. 2b). **b–d** ATAD5 (1602–1844) is a RBM. The indicated ATAD5 fragments were transiently expressed in 293T cells and immunoprecipitated using anti-FLAG beads. Bindings of RFC2–5 were monitored by co-immunoprecipitation of RFC4 or RFC5. **b** ATAD5 (1602–1719) is important for RFC2–5 binding. ATAD5 (1602–1844) bound to RFC2–5, but ATAD5 (1720–1844) did not. **c** ATAD5 (1720–1844) is also necessary for RFC2–5 binding. **d** E$^{1713}$ALSF$^{1717}$ of ATAD5 is important for RFC2–5 binding. The CM1 mutant, E$^{1713}$ALSF$^{1717}$ to AAGGG, was severely defective in RFC2–5 binding. **e–g** Proper RFC2–5 binding to ATAD5 is crucial for PCNA unloading. PCNA unloading activity of the indicated ATAD5 variants were examined as described in Fig. 1b. **e** RBM-deleted ATAD5 (693–1719) is defective in PCNA unloading. Graph shows relative amount of PCNA on the chromatin ($n = 3$). **f** ATAD5 (CM1), which is defective in RFC2–5 binding, is severely defective in PCNA unloading. Graph shows relative amount of PCNA on the chromatin ($n = 3$). **g** RBM-deleted ATAD5 (693–1719) fails to unload PCNA in vitro. In total, 1.4 kbps DNA was used for this assay. Graph shows relative PCNA amounts remained on DNA after unloading reaction ($n = 2$). See also Supplementary Fig. 2

they were not completely defective in RLC formation compared with the RBM-deletion mutant (ΔC1720) (Fig. 3c). Binding to RFC3, 4, and 5 was less affected by the EK mutation. Therefore, ATPase motifs of ATAD5 were not only essential for PCNA unloading, but they also influenced the stability of RLC.

We purified ATPase-motif mutants (KA, NQA, and EK) with RFC2–5 using Baculovirus expression system (Supplementary Fig. 3b). All mutants were purified as pentameric complexes. Consistent with cellular results in Fig. 3b, all ATPase-motif mutants were severely defective in PCNA unloading (Fig. 3d).

Next, we examined whether PCNA unloading affects DNA replication. We examined the S phase progression of cells expressing the unloading-defective EK mutant (Supplementary Fig. 3c and d). It has been previously suggested that PCNA unloading by ATAD5-RLC was important for proper S phase progression[3]. ATAD5 depletion resulted in markedly reduced EdU incorporation during the S phase. Wild-type ATAD5 expression in ATAD5-depleted cells restored EdU incorporation, but the EK mutant could not. Therefore, ATAD5-RLC-dependent PCNA unloading is important for proper DNA replication.

**ATAD5-RLC binding to PCNA triggers PCNA release from DNA.** Next, we investigated the molecular mechanism of PCNA loading and unloading. We performed ATP-γ-S experiments with ATAD5 (ΔN692)-RLC and RFC1 (ΔN554)-RFC purified from the Baculovirus expression system. In the case of RFC, ATP hydrolysis was suggested to be required for PCNA separation from the RFC-PCNA intermediate, followed by PCNA ring closure[37]. Supporting this model, ATP-γ-S, a non-hydrolysable analog of ATP, stalled the PCNA loading reaction at the DNA-binding step, and RFC-PCNA intermediates accumulated on DNA (Supplementary Fig. 3e). ATP-γ-S also interfered with efficient PCNA unloading by ATAD5-RLC (Fig. 3e (ATAD5 (ΔN692)-RLC purified from Baculovirus system) and Supplementary Fig. 3f (full-length ATAD5-RLC purified from the yeast-bacterial system)). However, at a higher concentration of ATAD5-RLC, PCNA was released from DNA. Assuming that ATP hydrolysis was required for the separation of AAA+ ATPase complex from its substrate, the released PCNA should be in the form of an ATAD5-RLC-bound intermediate. Therefore, the PCNA-ATAD5-RLC intermediate might have a conformation that facilitates its release from DNA.

To mechanistically analyze PCNA loading and unloading by RFC and ATAD5-RLC, we performed smFRET experiments using ATAD5 (ΔN692)-RLC and RFC1 (ΔN554)-RFC prepared from Baculovirus expression system (Fig. 4a). As PCNA forms a homo-trimeric ring, it is difficult to assess the geometry of the loaded PCNA unambiguously if a PCNA monomer within the trimer is specifically labeled. Thus, we fully labeled all three subunits with Cy3 at the two exposed cysteine residues (C62, C81), which allows for measuring the average distance between the PCNA ring and the label on the DNA without being obscured

by the variable orientation of PCNA trimer on DNA. Labeled PCNA proteins retained their loading activity (Supplementary Fig. 4a). The gapped DNA duplex was labeled with Cy5 on the template strand at the position of the 3′-end of the primer strand (Fig. 4a). One end of the DNA substrate was biotinylated for surface attachment and the other end had digoxigenin for blocking with anti-digoxigenin antibody to prevent the loaded PCNA from sliding off. PCNA loading was monitored by detecting PCNA spots on the surface that are co-localized with DNA spots. The changing number of PCNA spots during the loading and unloading reactions by RFC and ATAD5-RLC, respectively, confirmed the functionality of our experimental setup (Supplementary Fig. 4b–d). RFC did not show significant PCNA unloading activity (Supplementary Fig. 4d). In addition, the NQA and EK mutants of ATAD5-RLC did not exhibit any unloading activity, consistent with the bulk measurements (Fig. 3 and Supplementary Fig. 4e).

PCNA loaded by RFC exhibited a stable low-FRET state ($E_{FRET} = 0.34$) with transient spikes in FRET level, representing the diffusion of PCNA along the DNA substrate (Supplementary Fig. 4f-g). Real-time observation of PCNA loading revealed FRET dynamics followed by stepwise photobleaching of Cy3 dyes (Supplementary Fig. 4h). smFRET traces showing the fluorescence level of six Cy3 dyes were selected and the time range prior to the photobleaching of Cy3 dyes was used for further analysis. Remarkably, PCNA loading was found to occur through three distinct stages (Fig. 4b). The first one was a short-lived mid-FRET state ($E_{FRET} = 0.48$; loading intermediate 1 (LI1)). This state soon transitioned to a longer-lived high-FRET state ($E_{FRET} = 0.62$; loading intermediate 2 (LI2)). Finally, it reached a low-FRET state ($E_{FRET} = 0.34$; loaded state (LS)), which matches the dominant state found on pre-loaded PCNA. By contrast, PCNA unloading by ATAD5-RLC occurred in two stages (Fig. 4c). LS converted to a high-FRET state ($E_{FRET} = 0.58$; unloading intermediate (UI)), followed by the disappearance of fluorescence signal that represents PCNA dissociation from the substrate. The disappearance of Cy3 signal in a single step indicates that PCNA trimer dissociates without breaking into monomers. As these FRET transitions were consistently observed in most loading and unloading traces, overlaying many smFRET traces synchronized at the moment of PCNA-DNA association or dissociation produced heat maps that clearly displayed the stepwise transitions (Fig. 4d, e). FRET population maps built from PCNA loading traces with fewer Cy3 dyes exhibited broader FRET distribution but the major FRET levels were same as those from traces with six Cy3 dyes, confirming that the FRET signal from our multi-donor labeling scheme can be interpreted as usual with single donor–acceptor pair (Supplementary Fig. 4i). When Cy5 was placed at the 5′-end of the upstream primer or the 5′-end of the downstream primer, PCNA loading occurred with the same rate, and the FRET transition showed a similar pattern, confirming that observed traces do not represent artifactual dynamics due to

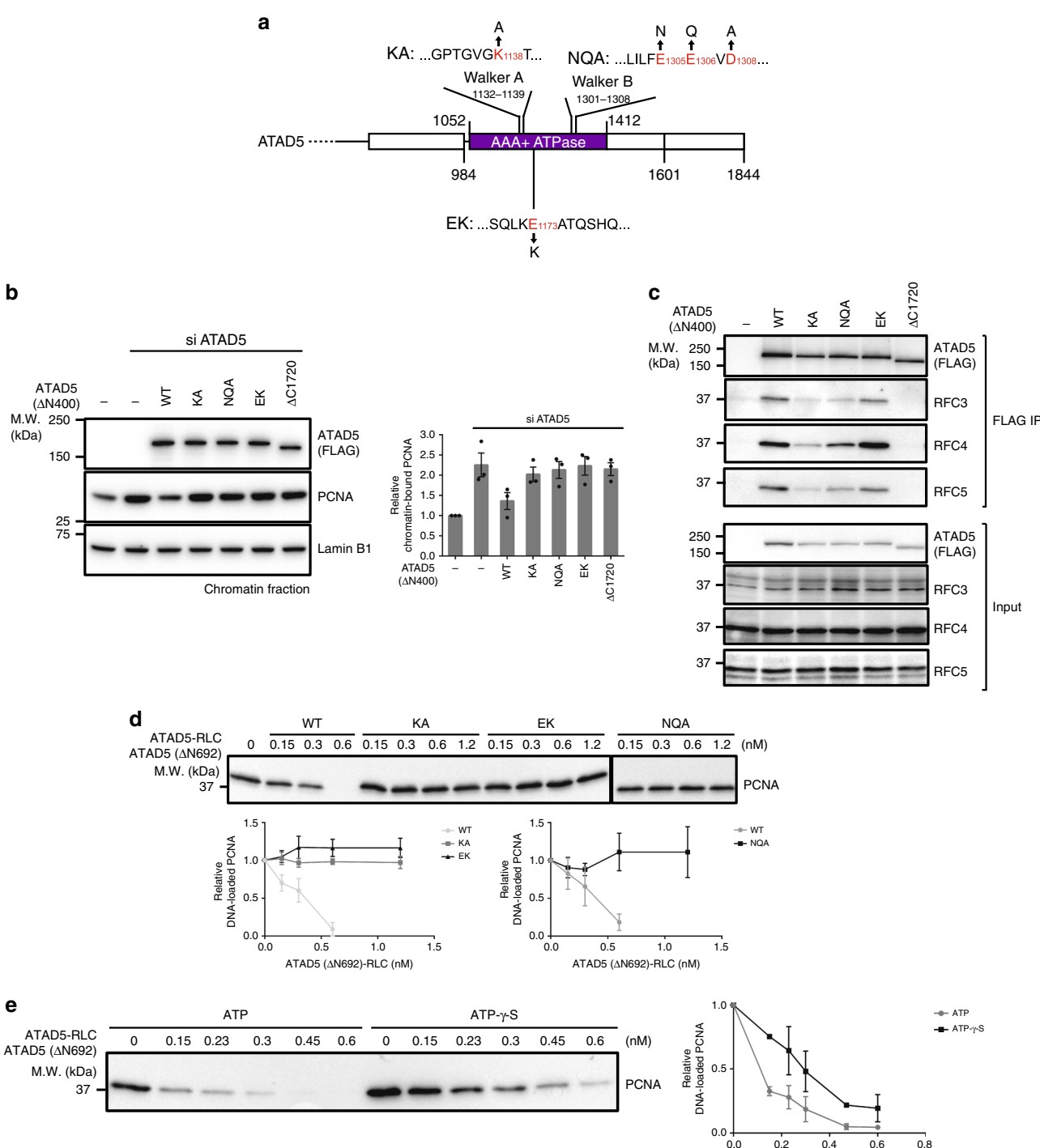

**Fig. 3** ATPase interface of ATAD5 is crucial for PCNA unloading. **a** ATPase-motif mutants examined in this study. Conserved lysine (K1138) in the Walker A motif or negatively charged amino acids in the putative Walker B motif (E1305, E1306, and D1308) were mutated as indicated (denoted as KA and NQA, respectively). E1173K mutation (EK) is located in the conserved region between Walker A and Walker B motifs. **b** ATPase-motif mutants are defective in PCNA unloading. ATAD5 (ΔN400) harboring mutations in the ATPase-motif were expressed in ATAD5-depleted 293T cells. Graph shows relative amount of PCNA on the chromatin (n = 3). The amount of PCNA on the chromatin shows that all ATPase-motif mutants are defective in PCNA unloading, similar to the RBM-deleted mutant (ATAD5 (ΔC1720)). **c** Functional Walker A and Walker B motifs are required for stable interaction between ATAD5 and RFC2–5. The indicated ATAD5 variants were affinity-purified and co-immunoprecipitated RFC3–5 were examined. The KA and NQA mutants show reduced RFC2–5 binding compared with wild-type and EK mutant. **d** ATPase-motif mutants of ATAD5-RLC fail to unload PCNA in vitro. Unloading activity of purified wild-type or mutant ATAD5-RLC was examined. In total, 1.4 kbps DNA was used for this assay. Graph shows relative PCNA amounts remained on DNA after unloading reaction (n = 3 for KA and EK experiments and n = 2 for NQA experiments. NQA experiments share control reaction with ATAD5 (693–1719) experiments (Fig. 2g), because two experiments were performed together). **e** ATP-γ-S partially reduces the efficiency of PCNA unloading by ATAD5-RLC. The PCNA unloading reaction was performed in the presence of ATP or ATP-γ-S. In total, 1.4 kbps DNA was used for this assay. Graph shows relative PCNA amounts remained on DNA after unloading reaction (n = 2). See also Supplementary Fig. 3

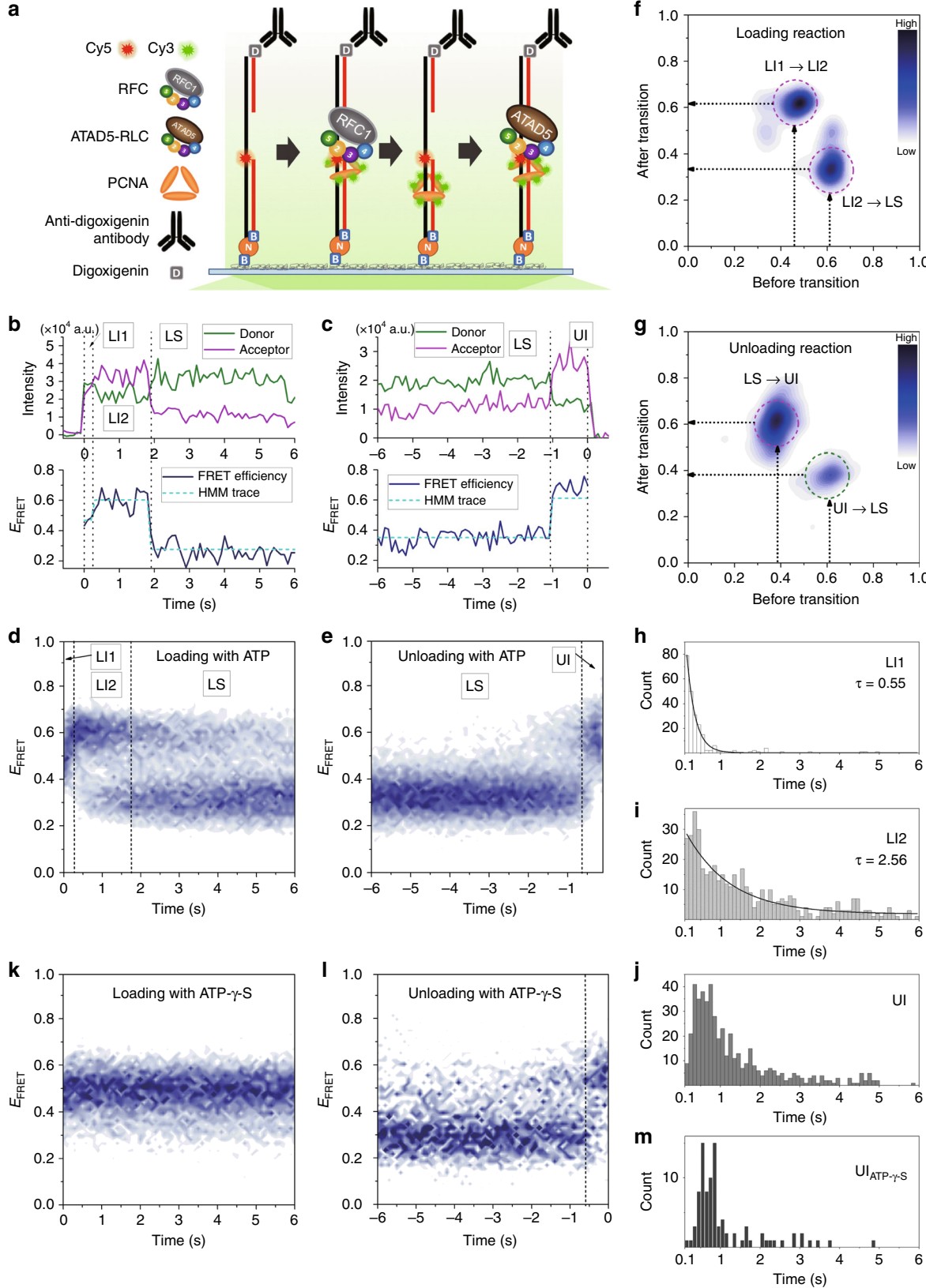

DNA labeling (Supplementary Fig. 4c, j–m). When Cy5 was placed at the 5′-end of the upstream primer, the FRET efficiency remained nearly zero, implying that PCNA remained mostly at the primer-template junction without frequent diffusion into duplex DNA (Supplementary Fig. 4j–k).

Hidden Markov analysis was successfully employed to distinguish the above-described FRET states and revealed their kinetics (Fig. 4f–j). A transition density plot (TDP) obtained from the first 6 s after PCNA-DNA association showed a high frequency of LI1→LI2 and LI2→LS transitions with much lower

**Fig. 4** Loading and unloading of PCNA occur through distinct steps. **a** Schematic diagram of smFRET measurements by total internal reflection microscopy to monitor PCNA loading and unloading. Cy3-labeled PCNA was pre-assembled with RFC and loaded to Cy5-labeled primed-template DNA, which was tethered to the surface via biotin (B) to neutravidin (N) binding, and the fluorescence signals were recorded. When loading was complete, ATAD5-RLC was added to observe the PCNA unloading process. **b** A representative smFRET trace from the PCNA loading experiment, showing Cy3 (green)/Cy5 (magenta) intensities, FRET efficiency (navy), and state sequence found from hidden Markov modeling (light blue). The association moment is defined as 0 s. **c** A representative smFRET trace from the PCNA unloading experiment. The dissociation moment is defined as 0 s. **d** Heat map of FRET population dynamics generated from 393 loading traces synchronized at PCNA-DNA association. Three distinct stages, LI1, LI2, and LS, are marked. **e** Heat map generated from 513 unloading traces synchronized at PCNA-DNA dissociation. Two distinct stages, LS and UI, are marked. **f** Transition density plot (TDP) generated from hidden Markov modeling on the first 6 s of the loading traces since PCNA-DNA association. LI1→LI2 and LI2→LS transitions are marked in red circles. **g** TDP generated from hidden Markov modeling on the last 6 s of the unloading traces prior to PCNA-DNA dissociation. LS→UI and UI→LS transitions are marked in red and green circles, respectively. **h–j** Dwell time distributions of LI1 (H), LI2 (I), and UI (J) states during PCNA loading and unloading processes. Dwell times of LI1 and LI2 were fitted to single exponential decay curve, $Ae^{-t/\tau}$, and $\tau$ values are shown. **k** Heat map generated from 230 loading traces with ATP-γ-S, showing the complex stalled at the initial FRET level. **l** Heat map generated from 102 unloading traces with ATP-γ-S showing similar transition to that with ATP. **m** Dwell time distribution of UI state measured with ATP-γ-S. See also Supplementary Fig. 4

frequency of the reverse transitions (Fig. 4f). TDP from the last 6 s before PCNA-DNA dissociation showed a high frequency of LS→UI transition, consistent with the synchronized heat map (Fig. 4g). There also existed less frequent reverse transitions (Fig. 4g, green circle). These reverse transitions represent occasional unloading trials that returned to the loaded state. Analysis of dwell time at each FRET state during PCNA loading revealed nearly exponential distributions for LI1 and LI2 (Fig. 4h, i), suggesting that each state transitioned mainly through a single rate-limiting step. By contrast, the dwell time distribution of UI was far from being exponential and showed a peak at non-zero time (Fig. 4j), which suggest the presence of multiple catalytic steps within UI during the unloading process.

To evaluate the role of ATP hydrolysis in PCNA loading and unloading processes, we used ATP-γ-S in single molecule measurement. Upon loading PCNA with ATP-γ-S, the FRET was stalled at a level matching that of LI1 (Fig. 4k). This indicates that LI1 and LI2 are two intermediate states separated by ATP hydrolysis, consistent with what was suggested from stopped-flow FRET measurements[20]. Previous studies have shown that the PCNA ring closes after ATP hydrolysis during loading. Thus, we designated LI1 as an open PCNA-RFC intermediate that initially bound to DNA. LS should represent the fully loaded PCNA after RFC release. Therefore, LI2 would represent an intermediate of closed PCNA before RFC release. By contrast, PCNA unloading by ATAD5-RLC still occurred using ATP-γ-S, as in the bulk reaction (Supplementary Fig. 4d). The unloading traces went through the same high-FRET state (UI) as those with ATP (Fig. 4l). Furthermore, the dwell time distribution of UI showed a peak at similar time for ATP and ATP-γ-S (Fig. 4j, m). These results show that PCNA unloading by ATAD5-RLC does not require ATP hydrolysis. ATP hydrolysis might rather induce the dissociation of ATAD5-RLC from PCNA after its release from DNA, to recycle both ATAD5-RLC and PCNA.

**ATAD5-RLC unloads ubiquitinated PCNA**. When replication forks encounter DNA lesions, PCNA is ubiquitinated. Bulky modifications like ubiquitination might interfere with PCNA unloading process. We therefore examined whether ATAD5 could unload Ub-PCNA. We performed Ub-PCNA unloading assay with ATAD5 (ΔN692)-RLC purified from the Baculovirus expression system or full-length ATAD5-RLC purified from the yeast-bacterial system. DNA-loaded PCNA was mono- or poly-ubiquitinated with purified ubiquitinating enzymes (UBA1, RAD6B-RAD18, UBC13-MMS2, and HLTF), and treated with ATAD5-RLC (Fig. 5a). UBA1 and the RAD6B-RAD18 complex efficiently mono-ubiquitinated PCNA on DNA (Fig. 5b and Supplementary Fig. 5a) compared with DNA-unbound PCNA. This result is consistent with previous reports showing that

RAD6B-RAD18 mono-ubiquitinates DNA-bound PCNA at stalled replication forks[33].

Remarkably, ATAD5 (ΔN692)-RLC efficiently unloaded mono-ubiquitinated PCNA (Fig. 5c, d, and Supplementary Fig. 5b). Unloading efficiency of full-length ATAD5-RLC was not significantly affected by PCNA mono-ubiquitination (Supplementary Fig. 5c).

Next, we examined the unloading of poly-ubiquitinated PCNA. PCNA was first mono-ubiquitinated with RAD6B-RAD18, then poly-ubiquitinated with UBC13-MMS2 and HLTF (Fig. 5e and Supplementary Fig. 5d)[32]. Poly-ubiquitination of PCNA was indicated by a ladder pattern of Ub-PCNA bands. These sequential two-step poly-ubiquitination reactions were more efficient than the one-step poly-ubiquitination reaction (compare lanes 3 and 4, Fig. 5e). After poly-ubiquitination, ATAD5 (ΔN692)-RLC or ATAD5-RLC was treated (Fig. 5f and Supplementary Fig. 5e). Both mono- and poly-ubiquitinated PCNA were released from DNA by ATAD5-RLC with similar efficiencies. In support of these in vitro results, ATAD5 (ΔN692) significantly reduced the chromatin-bound mono-ubiquitinated PCNA in ATAD5-depleted cells (Supplementary Fig. 5f). Therefore, ATAD5-RLC is a robust PCNA unloader that is able to unload both unmodified and ubiquitinated PCNA from DNA.

**Replication proteins regulates PCNA unloading in vitro**. We hypothesized that there might be a mechanism to prevent premature unloading of PCNA during DNA replication. DNA structures did not significantly affect PCNA unloading by ATAD5-RLC (Supplementary Fig. 1h–j). Another possibility is that replication enzymes might affect PCNA unloading during DNA replication. To examine this possibility, we took advantage of budding yeast replication proteins.

We first examined the activity of Elg1-RLC, a yeast homolog of ATAD5-RLC. Purified pentameric Elg1-RLC showed PCNA unloading activity (Fig. 6a and Supplementary Fig. 6a). Elg1 could unload PCNA in the presence of ATP-γ-S like ATAD5-RLC. Like human counterparts, only Elg1-RLC unloaded PCNA among RFC and three RLCs under our experimental setup (Fig. 6b, and Supplementary Fig. 6b). Like human CTF18-RLC, yeast Ctf18-RLC exhibited PCNA loading activity, although weaker than RFC (Supplementary Fig. 6c). Rad24-RLC loaded Ddc1-Mec3-Rad17 (yeast 9-1-1 complex) onto the DNA (Supplementary Fig. 6d). RPA enhanced the loading of Ddc1-Mec3-Rad17 by Rad24-RLC. Elg1-RLC did not affect the amount of yeast 9-1-1 complex remaining on DNA (Supplementary Fig. 6e). Therefore, Elg1-RLC is the most potent PCNA unloader among yeast clamp-loader complexes.

Elg1-RLC and ATAD5-RLC have very similar biochemical properties. In the ATPase motif of Elg1, Walker A

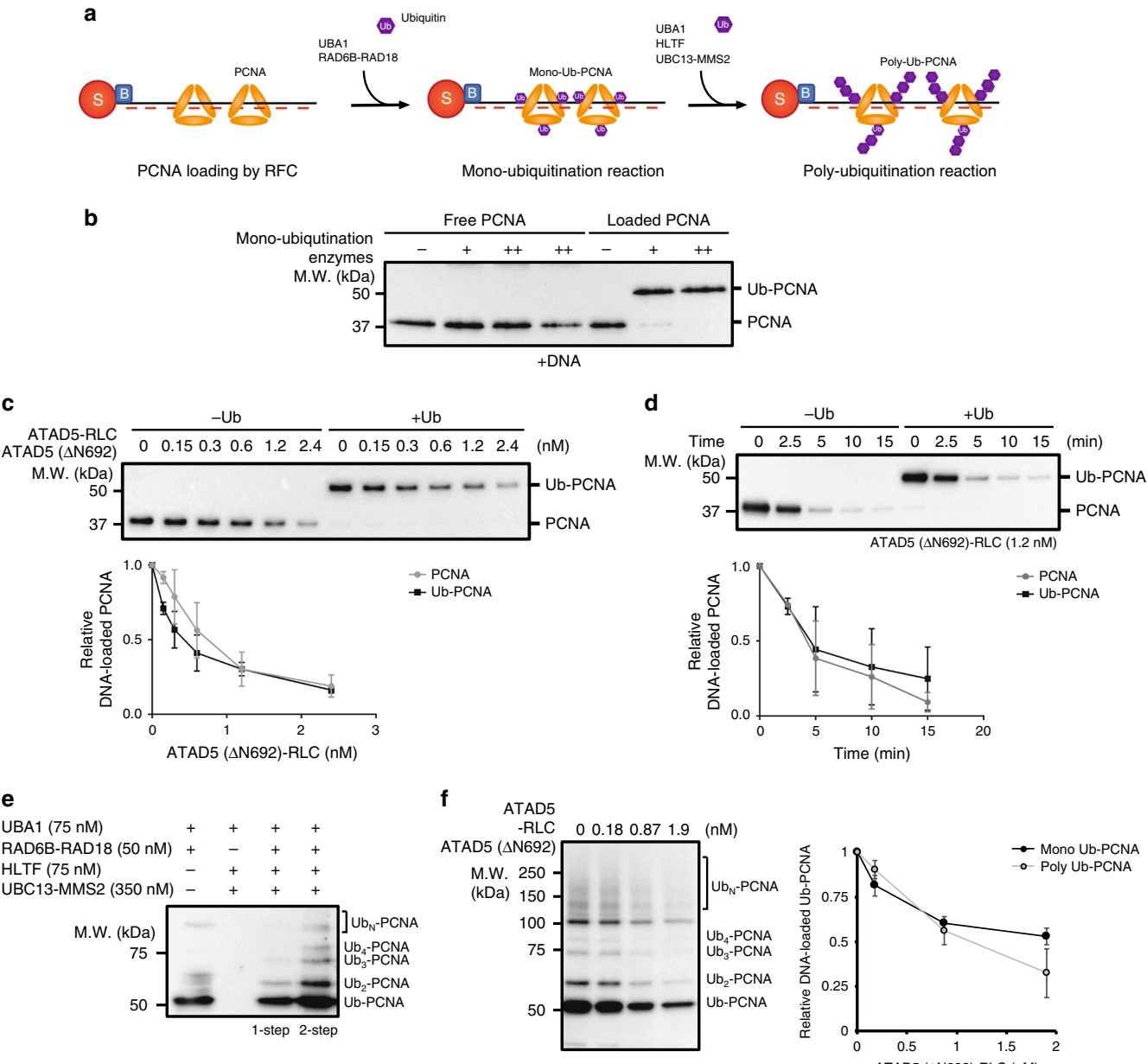

**Fig. 5** ATAD5-RLC unloads ubiquitinated PCNA. **a** Schematic diagram of PCNA mono- and poly-ubiquitination. After loading onto bead-coupled DNA, PCNA was mono-ubiquitinated with UBA1, RAD6B-RAD18, and ubiquitin. Mono-ubiquitinated PCNA was poly-ubiquitinated with UBA1, HLTF, UBC13-MMS2, and ubiquitin. DNA beads were washed after each reaction to remove ubiquitination enzymes and free ubiquitin. Unloading of mono-ubiquitinated PCNA (Ub-PCNA) or poly-ubiquitinated PCNA ($Ub_N$-PCNA) was monitored by immunoblot using anti-PCNA antibody unless indicated. **b** Mono-ubiquitination of PCNA. Same amount of free PCNA or DNA-loaded PCNA were ubiquitinated with mono-ubiquitination enzymes (+: 16.7 nM UBA1, 66.7 nM RAD6B-RAD18, ++: 33.4 nM UBA1, 133.4 nM RAD6B-RAD18). +DNA is a reaction in which free PCNA was mixed with bead-coupled DNA without loading reaction. DNA-loaded PCNA was preferentially mono-ubiquitinated. **c–d** ATAD5 (ΔN692)-RLC unloads mono-ubiquitinated PCNA. The indicated amount of ATAD5 (ΔN692)-RLC was used to treat DNA-loaded unmodified or mono-ubiquitinated PCNA. Graph shows relative PCNA or Ub-PCNA amounts remained on DNA after unloading reaction. In total, 1.4 kbps DNA was used for this assay. **c** ATAD5 (ΔN692)-RLC were titrated in the unloading reaction of PCNA or mono-ubiquitinated PCNA ($n = 3$). **d** Time-course unloading of PCNA or mono-ubiquitinated PCNA by 1.2 nM ATAD5 (ΔN692)-RLC ($n = 2$). **e** In vitro PCNA poly-ubiquitination. PCNA was loaded to 130-mer substrate DNA, then poly-ubiquitinated with the indicated enzymes. Poly-ubiquitination reaction was performed through a single step (1-step) or two steps (2-step). In two-step reactions, the poly-ubiquitination reaction was separately performed after the mono-ubiquitination reaction. The two-step reaction was more efficient compared with the single-step reaction. **f** ATAD5 (ΔN692)-RLC unloads mono- and poly-ubiquitinated PCNA with similar efficiency. After the mono- or poly-ubiquitination reaction, ATAD5 (ΔN692)-RLC was treated to DNA-loaded ubiquitinated PCNA. Unloading of ubiquitinated PCNA was monitored with anti-Ub-PCNA antibody. Graph shows relative mono-Ub-PCNA or poly-Ub-PCNA amounts remained on DNA after unloading reaction ($n = 3$). See also Supplementary Fig. 5

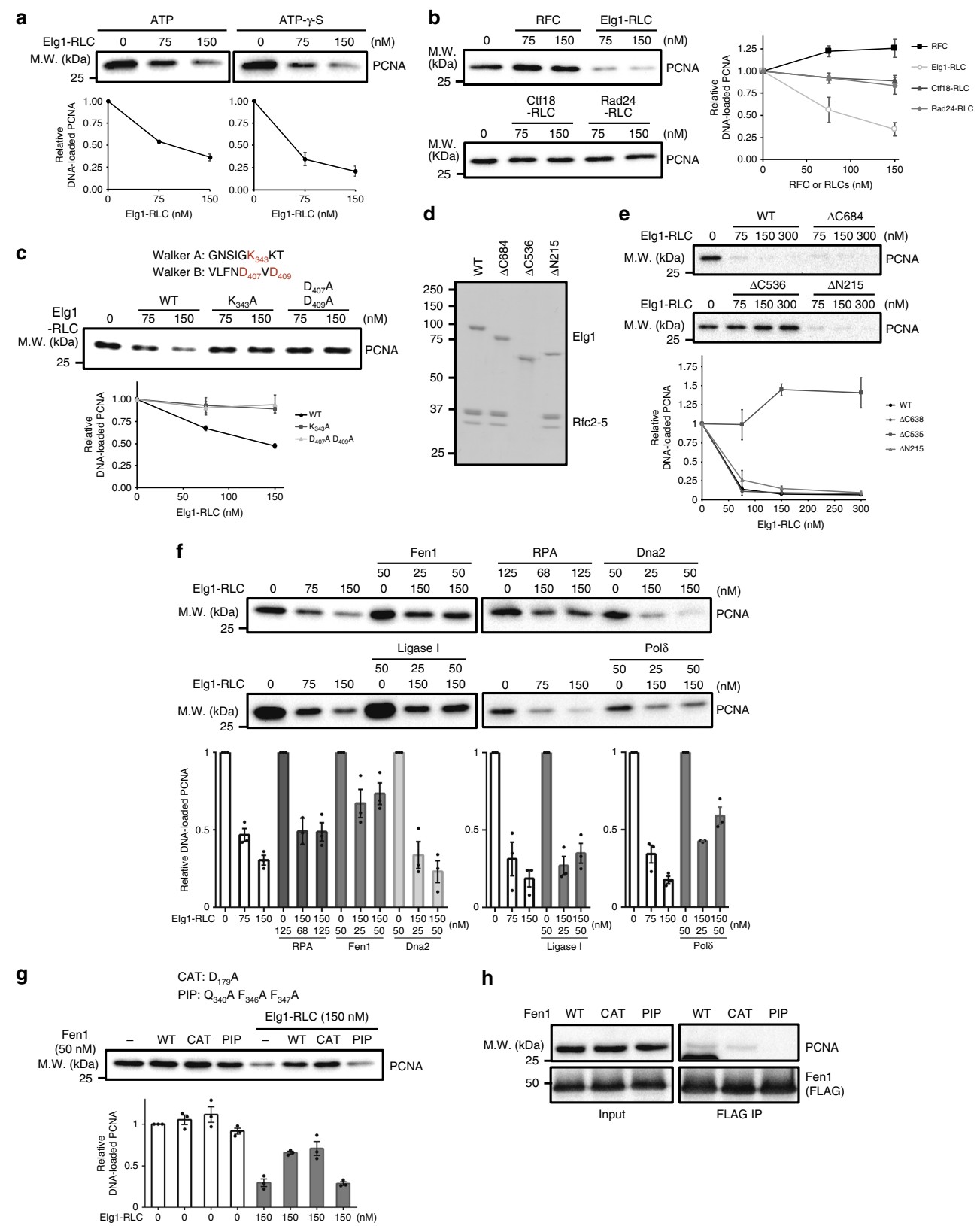

(GNSIGK$^{343}$KT) and Walker B (VLFND$^{407}$VD$^{409}$) motifs were conserved (Fig. 6c). A mutation in the conserved lysine (K343) in Walker A or mutations of two aspartates (D407 and D409) in Walker B abolished PCNA unloading activity (Fig. 6c and Supplementary Fig. 6f). Next, we examined PCNA unloading activity of Egl1 deletion mutants (Fig. 6d). Deletion of N-terminal 215 amino acids of Elg1 did not affect PCNA unloading activity of Elg1-RLC (Fig. 6e). Therefore, although Elg1-RLC is less

**Fig. 6** Okazaki fragment-processing enzymes inhibit PCNA unloading by Elg1-RLC. **a** Elg1-RLC unloads PCNA in vitro. DNA-loaded yeast PCNA (Pol30p) was treated with indicated amount of purified Elg1-RLC in the presence of ATP or ATP-γ-S. In total, 1.4 kbps DNA was used for PCNA unloading assays shown in this figure. Graphs show relative PCNA amounts remained on DNA after unloading reaction ($n = 3$ unless indicated). Error bar indicates standard deviation. **b** Elg1-RLC unloads PCNA. RFC, Ctf18-RLC, and Rad24-RLC did not unload PCNA from DNA. **c** The ATPase motif of Elg1-RLC is crucial for PCNA unloading activity. Mutation of the conserved lysine in Walker A motif or mutations of two aspartates in Walker B motif abolished the unloading activity of Elg1-RLC. **d** Purification of Elg1 N-terminal or C-terminal deletion mutants. Elg1 (ΔC536) failed to interact with Rfc2–5. **e** Elg1 (216-683) is crucial for PCNA unloading. PCNA unloading activity of indicated deletion mutants of Elg1-RLC were examined ($n = 2$). **f** Okazaki fragment-processing enzymes inhibit Elg1-RLC mediated PCNA unloading. Indicated proteins were added to the PCNA unloading reaction with Elg1-RLC. **g, h** PCNA-Fen1 interaction is required for the inhibition of Elg1-RLC mediated PCNA unloading. **g** PIP box mutant of Fen1 (PIP) did not inhibit PCNA unloading compared with the wild-type or nuclease-motif mutant (CAT). **h** PIP box mutant of Fen1 failed to interact with PCNA in vitro. Purified wild-type or mutant Fen1 were mixed with PCNA, and Fen1 was affinity-purified from the mixture using anti-FLAG beads. Co-isolated PCNA was analyzed by immunoblotting. See also Supplementary Fig. 6

efficient compared with human ATAD5-RLC, it is not due to the presence of N-terminal domain (Fig. 1f and Fig. 6a). Elg1 (ΔC536)-RLC did not bind to Rfc2–5 and was defective in PCNA unloading. These results suggest that the 215th to 683rd amino-acid region of Elg1 is crucial for PCNA unloading activity.

We examined whether ubiquitination or SUMOylation of PCNA could affect unloading by Elg1-RLC. PCNA was mono-ubiquitinated or SUMOylated with purified enzymes (Supplementary Fig. 6g). PCNA unloading by Elg1-RLC was not affected by ubiquitination or SUMOylation (Supplementary Fig. 6h).

To understand the effect of DNA-replication proteins on PCNA unloading, we added purified proteins to PCNA unloading reaction. We tested enzymes involved in Okazaki fragment maturation, such as RPA, Ligase I, Fen1, Dna2, and polymerase δ (Fig. 6f and Supplementary Fig. 6i). Among these, Fen1, which cleaves flaps during the maturation of Okazaki fragments, strongly inhibited the unloading reaction. RPA, Ligase I and polymerase δ also partially inhibited PCNA unloading. Dna2, which does not physically interact with PCNA, did not affect PCNA unloading activity of Elg1-RLC. Fen1 interacts with PCNA through the PCNA-interacting-protein (PIP) box. A PIP box mutant of Fen1 did not bind to PCNA and was much less effective in the inhibition of PCNA unloading (Fig. 6g, h). By contrast, a mutation in the nuclease motif of Fen1, Fen1 (CAT), still inhibited PCNA unloading. Therefore, our results suggest that replication proteins might inhibit PCNA unloading during DNA replication by preventing access of Elg1-RLC to PCNA.

## Discussion

Here, we demonstrate that ATAD5/Elg1-RLC functions as a molecular sweeper for PCNA remaining on DNA after the completion of DNA replication and repair.

There are structural and functional similarities between RFC and ATAD5-RLC. However, RFC and ATAD5-RLC mainly catalyzed reactions in opposite directions. Although RFC could reduce the amount of DNA-loaded PCNA, its PCNA unloading activity is significantly lower than that of ATAD5-RLC (Supplementary Fig. 1k–n). We speculate that the relative position of ATAD5 and RFC1 against RFC2–5 might be different in the complex, and such differences could determine the directionality of the reaction. The C-terminal RBM might be important for the positioning of ATAD5 and RFC1 in each complex. Supporting this idea, swapping the RBM of ATAD5 with that of RFC1 abolished PCNA unloading activity (Supplementary Fig. 2d, e).

The structural difference between ATAD5-RLC and RFC affected their DNA-binding affinity after PCNA association. In the case of RFC, RFC-PCNA intermediates binds to DNA, and this binding triggers ATP hydrolysis and PCNA loading[37]. ATP-γ-S stalled the RFC-PCNA intermediates on DNA (Supplementary Fig. 3e). Single molecule experiments with RFC revealed three conformational stages during PCNA loading (Fig. 4). Each

stage was shown to represent the initial binding of RFC-PCNA to DNA, ATP hydrolysis and closure of the PCNA ring, and the release of RFC. However, during PCNA unloading, ATP-γ-S did not inhibit the reaction. PCNA unloading occurred in two steps, possibly representing ATAD5-RLC binding to the DNA-loaded PCNA that resulted in PCNA ring opening and the release of ATAD5-RLC-PCNA intermediate from DNA. As the release of the complex occurred both with ATP and with ATP-γ-S, it is possible that the ATAD5-RLC-PCNA intermediate immediately takes a conformation having much lower affinity to DNA than that of RFC-PCNA complex. ATP hydrolysis might occur after or during dissociation from DNA, which could trigger the separation of ATAD5-RLC from PCNA to reset the complexes. ATPase-motif mutants might fail to reduce the level of chromatin-bound PCNA because mutant ATAD5-RLC cannot be recycled due to ATP binding/hydrolysis defect. Otherwise, ATPase-motif mutants might take a conformation that is defective in PCNA unloading. Although RFC could unload PCNA, its PCNA unloading activity is significantly lower than that of ATAD5-RLC (Supplementary Fig. 1k–n). Different from ATAD5-RLC, previous reports showed that RFC-mediated PCNA unloading requires ATP hydrolysis[10]. RFC-mediated PCNA unloading was affected by DNA structure, while ATAD5-RLC could unload PCNA regardless of DNA structure (Supplementary Fig. 1h–j, m). Our results provide a mechanistic basis to understand how two structurally related AAA+ ATPase complexes mainly catalyze the reaction in opposite directions.

The ubiquitination or SUMOylation site of PCNA is located on the back face of the PCNA trimer[38–40]. RFC binds to the front face of the PCNA trimer during PCNA loading[12]. It was reported that RFC also could unload mono-ubiquitinated PCNA[41]. The efficient unloading of ubiquitinated or SUMOylated PCNA by ATAD5/Elg1-RLC suggests that ATAD5/Elg1-RLC also accesses the DNA-bound PCNA from the front face of the PCNA trimer. Our results suggest that ATAD5-RLC might turn off Ub-PCNA-dependent lesion-bypass through unloading Ub-PCNA. We have previously shown that ATAD5 participates in de-ubiquitination of Ub-PCNA through its interaction with UAF1-USP1[30]. Why does human ATAD5 possess two inactivation mechanisms—unloading and de-ubiquitination—against Ub-PCNA? Ub-PCNA might be rapidly de-ubiquitinated during unloading by nearby de-ubiquitination enzymes. This coordinated de-ubiquitination and unloading processes might be important for efficient recycling of unmodified PCNA for new rounds of replicative DNA synthesis.

Although the difference was not significant, full-length ATAD5-RLC unloaded mono-Ub-PCNA slightly less efficiently compared with unmodified PCNA (Supplementary Fig. 5c). In the case of ATAD5 (ΔN692)-RLC, PCNA mono-ubiquitination did not affect PCNA unloading efficiency (Fig. 5c). ATP-γ-S less affect PCNA unloading activity of full-length ATAD5-RLC

compared with that of ATAD5 (ΔN692)-RLC (Fig. 3e and Supplementary Fig. 3f). Therefore, it is possible that N-terminal domain of ATAD5-RLC has a role in the regulation of PCNA unloading process.

Premature unloading of PCNA should be inhibited to prevent futile PCNA loading/unloading cycle. We found that yeast replication proteins, especially Fen1, could inhibit PCNA unloading by Elg1-RLC through its binding to PCNA in vitro (Fig. 6). A report have suggested that Fen1 could mask the front face of PCNA[42]. Thus, Fen1-PCNA interaction is likely to hinder the access of ATAD5-RLC to PCNA.

PCNA sequentially interacts with many replication enzymes during the replication process, especially for lagging strand synthesis. PCNA is initially loaded by RFC to the primer ends, and then pairs with DNA polymerase during DNA synthesis. PCNA recruits Fen1 and Ligase I during maturation process of Okazaki fragments. It has been previously reported that PCNA unloading by Elg1-RLC required the ligation of Okazaki fragments[29]. Competition between the above-mentioned PCNA-interacting proteins and ATAD5/Elg1-RLC for PCNA might suppress PCNA unloading until DNA synthesis is completed. It was reported that RFC-mediated PCNA unloading is also inhibited by polymerase δ[20]. Therefore, competition between clamp-loader complexes and PCNA-interacting proteins might serve as a mechanism for PCNA cycling. It is possible that other factors are involved in timely dissociation of PCNA by specifically recruiting ATAD5-RLC to completely replicated chromatin. For example, chromatin assembly factors may regulate the PCNA unloading process by modulating the interaction between ATAD5-RLC and DNA-loaded PCNA.

## Methods

**Protein purification**. Most of human proteins were expressed and were purified using the Bac-to-Bac Baculovirus expression system (Thermo Fisher Scientific). Viruses were prepared using Sf9 cells (Thermo Fisher Scientific), and proteins were expressed with Hi-5 cells (Thermo Fisher Scientific). Expression constructs for PCNA, RFC1, RFC1 (ΔN554), ATAD5 (ΔN692), and RAD17 contain N-terminal 10×HIS-tag and C-terminal 3×FLAG tag. CTF18 was tagged with N-terminal 10×HIS and C-terminal 2×StrepII-3×FLAG. RFC2–5 were cloned into the pFL multi-gene expression system and expressed from a single construct (see Supplementary Methods for primers). For purification of RFC and RLCs, viruses encoding RFC1, ATAD5, CTF18, or RAD17 were co-infected with a virus encoding RFC2–5. To purify RAD6B-RAD18, virus encoding 10×HIS-RAD18 −3×FLAG was co-transduced with RAD6B-encoding virus. Full-length human UBC13 (UBE2N) and MMS2 (UBE2V2) were cloned into pFL with N-terminal 10×His tag and C-terminal 3×Flag tag, respectively, and co-expressed in Hi-5 cells (see Supplementary Methods for primers). Full-length human HLTF was cloned into pFastBac1 plasmid with N-terminal 10×HIS tag and C-terminal 3×Flag tag, and expressed in Hi-5 cells. UBA1 was expressed in E. coli BL21 (DE3) cells (Enzynomics). Human ubiquitin was purchased from Sigma (U5382). Yeast RFC, Elg1-RLC, Ctf18-RLC, and Rad24-RLC were expressed and purified from yeast strains that harbor a Gal-inducible expression cassette for each complex (Supplementary Table 1). PCNA-ubiquitination or SUMOylation enzymes were purified from yeast strains containing a Gal-expression cassette for each gene. Yeast PCNA was expressed and purified from E. coli BL21 (DE3).

To purify proteins from insect cells, cell lysates were obtained by sonication and ultracentrifugation in lysis buffer (300 mM potassium acetate [KoAc] Buffer H [25 mM HEPES at pH 7.5], 1 mM EDTA, 1 mM EGTA, 2.5 mM magnesium acetate, 10% glycerol, 1 mM ATP, and 0.02% NP40, with 1× complete protease inhibitor cocktail [Roche]). Lysates were clarified by ultracentrifugation (36,000 × g, 60 min), and proteins were purified by sequential application of complete His Tag resin, anti-FLAG M2 agarose resin, and ion exchange chromatography. Purified proteins were analyzed by SDS-PAGE and Coomassie blue staining. Aliquoted proteins were frozen and stored at −80 °C.

To purify proteins from yeast cells, 2 L cultures were grown to O.D. = 0.8, and protein expression was induced by the addition of 2% galactose for 5 h. Cells were harvested and resuspended in 1/3 cell pellet volume of yeast lysis buffer (1.5 M potassium acetate, 0.8 M sorbitol, 10 mM magnesium acetate, 5% glycerol, 0.02% NP40) with 1× complete protease inhibitor cocktail. Resuspended cells were frozen by dripping into liquid nitrogen. The frozen cell drops were broken with a SPEX SamplePrep FreezerMill. After thawing, 15 mL of 0.3 M KoAc buffer H was added to the lysed cells. Cell lysates were cleared by ultracentrifugation (36,000 × g, 60

min) and proteins were purified by sequential application of anti-FLAG M2 agarose resin and ion exchange chromatography.

Wild-type human RFC, ATAD5-RLC or ATAD5 deletion mutants were also purified from the combination of yeast and bacterial expression system. RFC1 or ATAD5 was co-overexpressed in yeast cells with human RFC2–5 using a galactose-inducible system. Yeast cell lysates were prepared as described above. Because co-overexpression of RFC2–5 was not sufficient to obtain RFC or RLC, RFC2–5 were supplemented using the bacterial expression system. Human RFC2–5 were co-overexpressed in E. coli (BL21 (DE3)) harboring pDuet RFC2–5. E. coli cells were resuspended in 0.3 M KoAc Buffer H and lysed with lysozyme. Cell lysates were cleared by ultracentrifugation (36,000 × g, 60 min). Prepared yeast cell lysate was mixed with same volume of E. coli lysate and the mixed lysate was incubated for 8–12 h at 4 °C. RFC or ATAD5-RLC was purified by anti-FLAG agarose resin followed by SP Sepharose. For the purification of ATAD5 (ΔN692) alone, only ATAD5 (ΔN692) was overexpressed in yeast cells.

**DNA substrates for PCNA loading and unloading reaction**. The 1.4 kbps primer-template DNA was prepared as follows: 1.4 kbps dsDNA was prepared by PCR with biotinylated primer and pUC19 ARS1 (see supplementary Methods for primer information)[43]. Amplified DNA fragments were purified with 4 mL Centricon (10 kDa cut-off [Millipore]). Purified 1.4 kbps DNA was denatured and re-annealed in the presence of 20-fold molar excess of nine short primers (see supplementary Methods for primer information). Average distances between primers are ~150 bps. The annealed primer-template DNA was attached to streptavidin-coated magnetic beads (Dynabeads M-280 [Invitrogen]), and unbound DNA/oligonucleotides were washed out. Bead-attached DNA was resuspended in 0.3 M KoAc Buffer H without magnesium acetate. 130-mer DNA substrate was prepared by annealing two oligonucleotides to a 130-mer oligonucleotides (CTCTA TAAGA TATAG TCAAG TTCAG ACGTC CATGC CCTTA TCGGA GTCTC CGGCA AATGC AATGC TCAGC ATCTC CAGCC GCTTA GCATA CTTGC CGATG TACAT GAAAG CTTTG TCTGT CAGCA GGCCG) DNA. One oligonucleotide was a 5′ biotinylated 50-mer (CGGCC TGCTG ACAGA CAAAG CTTTC ATGTA CATCG GCAAG TATGC TAAGC) annealed to the 3′ portion of 130-mer oligonucleotides, and the other was a 70-mer (GCTGA GCATT GCATT TGCCG GAGAC TCCGA TAAGG GCATG GACGT CTGAA CTTGA CTATA TCTTA TAGAG) that was annealed to the 5′ portion of the 130-mer. The resulted DNA substrates contain a 10 nucleotides gap. To prepare nicked 130-mer DNA, 80-mer (GGCTG GAGAT GCTGA GCATT GCATT TGCCG GAGAC TCCGA TAAGG GCATG GACGT CTGAA CTTGA CTATA TCTTA TAGAG) was annealed instead of 70-mer. 5′ of 80-mer was phosphorylated. To ligate nicked DNA, T4 DNA ligase was treated after PCNA loading reaction (200 units of T4 DNA ligase in 0.3 M KoAc buffer H, 30 min at 37 °C, 1200 rpm on Thermomixer). Next, 80-mer DNA substrates were prepared by annealing two oligonucleotides to an 80-bp oligonucleotide (CTCTA TAAGA TATAG TCAAG TTCAG ACGTC TCGGC GTCTC CGGCC AATGC GTCTC CAGCC GCGCC GGCGC GCCGC CGACG, Supplementary Fig. 1h). One oligonucleotide was a 18-mer (CGTCG GCGGC CGCTC GGC) annealed to the 3′ portion of the 80-mer DNA, and the other was a 61-mer (CGGCT GGAGA CGCAT TGGCC GGAGA CGCCG AGACG TCTGA ACTTG ACTAT ATCTT ATAGA G) or 51-mer (CGCAT TGGCC GGAGA CGCCG AGACG TCTGA ACTTG ACTAT ATCTT ATAGA G) that was annealed to the 5′ portion of the 80-mer. The resulting two DNA substrates contain a 1 or 11 nucleotides gap, respectively. Each substrate has a TALE-binding sequence on the opposite side of biotinylation. Substrate DNA was attached to the magnetic beads and stored in 0.3 M KoAc Buffer H without magnesium acetate. In the case of short DNA substrate, 0.5 pmol DNA substrate was pre-incubated with 50 nM TALE in 40 μL reaction mixture (25 mM HEPES [pH 7.5], 300 mM KoAc, 1 mM EDTA, 1 mM EGTA, 2.5 mM magnesium acetate, 10% glycerol, 1 mM DTT, 1 mM ATP, 0.02% NP40) at 37 °C for 30 min before the PCNA loading reaction.

**PCNA loading and unloading reaction**. Two buffers were prepared for the PCNA loading/unloading reaction. A standard 2× loading reaction buffer (50 mM HEPES [pH 7.5], 24 mM magnesium acetate, 0.2 mM zinc acetate, 2 mM dithiothreitol [DTT], 40 mM phosphocreatine, 12 mM ATP, 0.04% NP40, 20% glycerol, 0.8 mg/mL bovine serum albumin, and 2× complete protease inhibitor cocktail [Roche]) and a protein dilution buffer (25 mM HEPES [pH 7.5], 300 mM KoAc, 1 mM EDTA, 1 mM EGTA, 2.5 mM magnesium acetate, 10% glycerol, 1 mM DTT, 1 mM ATP, and 0.02% NP40). Initially, 20 μL of 2× loading reaction buffer was mixed with 20 μL of protein mix that contains RFC (12.5 nM or indicated amount) and PCNA (250 nM). The PCNA loading reaction was performed by adding 40 μL of the reaction mixture to bead-conjugated DNA substrate. The reaction mixture was incubated in a Thermomixer (Eppendorf) for 30 min at 37 °C and 1200 rpm. After the reaction, the remaining RFC and unbound PCNA were removed by washing the beads once with 0.3 M KCl, twice with 0.5 M KCl, and once again with 0.3 M KoAc Buffer H. For the PCNA unloading reaction, 20 μL of 2× loading reaction buffer was mixed with 20 μL of protein mix that contains various amounts of ATAD5-RLC. The unloading reaction mixture was added to the collected DNA beads on which PCNA was loaded. The unloading reaction mixture was incubated in a Thermomixer for 15 min at 37 °C. Afterward, the reaction beads were washed with 0.3 M KCl Buffer H three times. Finally, DNA was resuspended in 15 μL of digestion buffer (25 mM Tris–HCl [pH 7.5], 100 mM KoAc, 5 mM CaCl$_2$, 5 mM

$MgCl_2$) containing 1 unit of DNase I (Promega). DNA was digested for 10 min, beads were collected using a magnet, and supernatants were taken. PCNA level on DNA was analyzed by immunoblotting. For the ATP-γ-S experiment, ATP-γ-S was added to the 2× loading reaction buffer and protein dilution buffer instead of ATP. For the loading/unloading reaction of yeast PCNA, reaction procedures were the same except that the reaction temperature was 30 °C and unloading reaction time was 5 min. After loading yeast PCNA, DNA beads were washed once with 0.5 M KCl buffer H and once with 0.1 M KoAc Buffer H. After unloading reaction, DNA beads were washed once with 0.3 M KoAc Buffer H and once with 0.1 M KoAc buffer H before DNase I digestion. For the loading of human 9-1-1 complex, RAD9-HUS1-RAD1 and RAD17-RLC were added to the reaction instead of PCNA and RFC, respectively. The yeast 9-1-1 loading reaction was performed with Ddc1-Mec3-Rad17 and Rad24.

**Mono-ubiquitination reactions.** For the mono-ubiquitination reaction, a standard 4× ubiquitination reaction buffer was prepared (80 mM HEPES [pH 7.5], 4 mM DTT, 40 mM magnesium chloride, 4 mM ATP). A mono-ubiquitination reaction mixture containing 75 nM UBA1 and 50 nM RAD6B-RAD18 was prepared by mixing 7.5 µL of 4× ubiquitination reaction buffer with 22.5 µL of protein mixture (1× mixture: 16.7 nM UBA1, 66.7 nM RAD6B-RAD18, and 0.83 µM ubiquitin) in the protein dilution buffer. Next, 30 µL of the mono-ubiquitination reaction buffer was added to the collected DNA beads on which PCNA was loaded. After incubating the reaction mixture on a Thermomixer (30 °C, 1200 rpm) for 1 h, the reaction was stopped by adding 25 nM EDTA. Beads were sequentially washed with 0.5 M KCl Buffer H and 0.3 M KCl Buffer H before additional treatments.

**Poly-ubiquitination reactions.** Poly-ubiquitination was performed after mono-ubiquitination reaction. After the mono-ubiquitination reaction, mono-ubiquitination enzymes were removed by washing magnetic beads with 0.5 M KCl and 0.3 M KCl Buffer H, sequentially. Next, 75 nM of HLTF and 350 nM of UBC13/MMS2 in 1× 30 µL ubiquitination reaction buffer were added to the bead and incubated for 30 min at 30 °C in a Thermomixer. The poly-ubiquitination reaction was stopped on ice, and beads were sequentially washed with 500 mM KCl Buffer H and 300 mM KCl Buffer H before additional treatments.

**Mono-ubiquitination and SUMOylation of yeast PCNA.** Mono-ubiquitination of SUMOylation of yeast PCNA was performed with PCNA loading reaction. For mono-ubiquitination, 130 nM of Uba1, 230 nM of Rad6-Rad18, and 2.5 µM of recombinant yeast ubiquitin (Abcam) were added to PCNA loading reaction. Reaction mixtures were incubated in Thermomixer (30 °C, 60 min, 1200 rpm). For SUMOylation, 100 nM Uba2-Aos1, 110 nM Ubc9, 200 nM Siz1, and 3.5 µM Smt3 were added to PCNA loading reaction. Reaction mixtures were incubated at 30 °C for 60 min in a Thermomixer (1200 rpm).

**Plasmids (or) DNA constructs and siRNAs.** Plasmids expressing full-length wild-type ATAD5 or its mutants were cloned into p3×FLAG-CMV10 expression vector (Sigma-Aldrich) or pcDNA™5/FRT/TO (Thermo Fisher Scientific) (see Supplementary Table 2). Site-directed mutagenesis was performed using the QuikChange site-directed mutagenesis kit (Agilent Technologies) according to the manufacturer's instructions to generate plasmid DNA for ATAD5 K1138A, E1173K, E1305NE1306QD1308A, CM1, CM2, CM3, and CM4 mutants. All constructs were confirmed by sequencing.

**siRNAs.** The following siRNAs against the 3′ untranslated region (UTR) of ATAD5 was synthesized from Bioneer (South Korea): 5′-GGAAGGUAGAGUU-CAUUAAUU-3′ (sense) and 5′-UUCCUUCCAUCUCAAGUAAUU-3′ (antisense)[3]. Non-targeting control siRNA (AccuTarget Negative Control siRNA) was purchased from Bioneer (South Korea).

**Cell culture.** Human embryonic kidney (HEK) 293T cells, U2OS cells (purchased from American Type Culture Collection (Manassas, VA)) and their derivative cell lines stably expressing ATAD5 protein were cultured in Dulbecco's modified Eagle's medium (Hyclone) supplemented with 10% fetal bovine serum (Hyclone) and 1% penicillin-streptomycin (Gibco) at 37 °C under 5% $CO_2$.

**Antibodies.** The following antibodies were used: anti-PCNA (PC10) antibody (Santa Cruz Biotechnology, sc-56, 1:2000); anti-Ubiquityl-PCNA (Lys164) antibody (Cell Signaling, #13439, 1:1000); anti-FLAG antibody (Sigma, F3165, 1:1000); anti-V5 antibody (Sigma, V8137, 1:2000); anti-RFC3 antibody (Bethyl, A300-186A, 1:500); anti-RFC4 antibody (Abcam, ab182145, 1:1000); anti-RFC5 antibody (Abcam, ab79871, 1:500); anti-Lamin B1 antibody (Abcam, ab16048, 1:1000); anti-Histone H3 antibody (Merck-millipore, 07-690, 1:5000); anti-Strep-Tactin HRP Conjugate antibody (IBA-life sciences, 2-1502-001, 1:1000). The anti-human ATAD5 antibody (1:1000) was raised in rabbits using the N-terminal 1–297 amino-acid fragment and then affinity-purified[3].

**Transfections and RNA interference.** Transfections of plasmid DNA and siRNAs were performed using X-tremeGENE HP DNA transfection Reagent (Roche) and Lipofectamine RNAiMAX (Invitrogen), respectively, according to the manufacturer's instructions. Transfected cells were harvested 48 h after transfection for further analysis. To deplete only endogenous ATAD5, siRNA targeting 3′UTR of ATAD5 was transfected 4 h after transfecting plasmids expressing wild-type or mutant ATAD5 protein.

**Chromatin fractionation.** Cells were lysed with buffer A (100 mM NaCl, 300 mM sucrose, 3 mM $MgCl_2$, 10 mM PIPES (pH 6.8), 1 mM EGTA, and 0.2% Triton X-100, containing the phosphatase inhibitor PhosSTOP (Roche) and complete protease inhibitor cocktail (Roche)) for 8 min on ice. Crude lysates were centrifuged at $5000 \times g$ at 4 °C for 5 min to separate the chromatin-containing pellet from the soluble fraction. The pellet was digested with 50 units of Benzonase (Enzynomics) for 40 min in RIPA buffer (50 mM Tris–HCl (pH 8.0), 150 mM NaCl, 5 mM EDTA, 1% Triton X-100, 0.1% SDS, 0.5% Na deoxycholate, 1 mM PMSF, 5 mM $MgCl_2$, containing the phosphatase inhibitor PhosSTOP (Roche) and complete protease inhibitor cocktail (Roche)) to extract chromatin-bound proteins. The chromatin-containing fractions were clarified by centrifugation ($13,000 \times g$, 4 °C) for 5 min to remove debris.

**Immunoprecipitation and western blot analysis.** Whole-cell lysates were prepared by lysing cells with buffer X (100 mM Tris–HCl (pH 8.5), 250 mM NaCl, 1 mM EDTA and 1% Nonidet P-40, 5 mM $MgCl_2$) supplemented with the phosphatase inhibitor PhosSTOP (Roche), complete protease inhibitor cocktail (Roche), and 500 units of Benzonase for 40 min at 4 °C. Lysates were cleared by centrifugation ($13,000 \times g$, 4 °C, 5 min). FLAG-tagged ATAD5 proteins were incubated with anti-FLAG M2 agarose affinity beads (Sigma) for 1 h at 4 °C with constant rotation. The beads were washed three times using buffer X, and the bead-bound proteins were eluted with buffer X containing 0.15 mg/mL FLAG peptide. Co-immunoprecipitated proteins were analyzed by immunoblotting. For V5 immunoprecipitation, V5-tagged proteins were isolated with anti-V5 agarose affinity gel (Sigma). Anti-V5 beads were resuspended in 1× SDS loading buffer.

**EdU incorporation analysis.** U2OS cells harboring doxycycline-inducible wild-type or E1173K mutant ATAD5 were labeled with 1 µM EdU. For flow cytometry analysis, samples were prepared using the Click-iT™ Plus EdU Alexa Fluor™ 647 Flow Cytometry Assay Kit (Invitrogen) according to the manufacturer's instructions and subjected to FACS analysis (FACSVerse flow cytometer, BD Biosciences). Data were analyzed using the FlowJo software (Tree Star).

**DNA template synthesis for single molecule measurements.** Oligonucleotides were synthesized by IDT with the following sequence design (See Supplementary Methods for information about oligonucleotides sequences). Strand 1 was the template strand, and strand 2 was the primer strand with the biotin modification at the 5′ end for surface immobilization. Strand 3 has a digoxigenin at the 3′ end for binding anti-digoxigenin antibody and physically blocking PCNA from sliding off the DNA. Amine-modified DNAs were labeled with Cy5 NHS ester (Lumiprobe, 20320) with final concentrations of 0.5 mM DNA and 5 mM Cy5 NHS ester in 100 mM sodium tetraborate for 6 h at room temperature, being gently shaken in dark. Labeled DNA oligonucleotides were recovered by ethanol precipitation and the labeling efficiency was 75–85%. Three strands were annealed at 10 µM each in 10 mM Tris–HCl pH 7.5, 50 mM NaCl, 1 mM EDTA to construct the primer-template DNA.

**PCNA purification and labeling.** For smFRET measurements, we labeled two surface-exposed cysteine residues (residues C62, C81, PBD ID: 1AXC). Cy3-maleimide (GE Healthcare, PA13130) at 100-fold excess to the target cysteine residues was added to PCNA protein at 33 µM monomer concentration and incubated at room temperature for 2 h with 10-fold excess of TCEP. Labeled PCNA was separated from free dyes by P6 size exclusion column (Bio-Rad, 732-6200). Labeling efficiency was calculated from Cy3 concentration measured by absorption at 550 nm ($150,000\ M^{-1}\ cm^{-1}$) and PCNA concentration measured by Bradford assay to show that each PCNA monomer contained 1.2 Cy3 molecules on average.

**smFRET measurement.** DNA substrate was immobilized on a PEG-coated quartz slide through biotin-neutravidin interaction as previous described[44]. Imaging buffer contained 50 mM HEPES pH 7.5, 14.5 mM Mg(oAc)₂, 0.1 mM ZnoAc, 1.5 mM DTT, 6.5 mM ATP, 0.03% NP40, 20% Glycerol, 0.4 mg/ml BSA, 1 mM EDTA, 1 mM EGTA, 150 mM KoAc, 1 mg/ml glucose oxidase, 0.04 mg/ml catalase, 0.6% glucose, and 3 mM Trolox. For equilibrium measurements, the pre-incubated RFC-PCNA complex was added to the surface at 1 nM in the imaging buffer. For flow-in measurements, RFC-PCNA (1 nM) or ATAD5 complexes (10 nM) were flowed in using a syringe pump for 1 s at 1.2 mL/min, 5 s after recording started. Home-built total internal reflection fluorescence microscope with an EMCCD camera (ANDOR iXon Ultra 897), constructed similar to one previously described[45] with two channels for Cy3 and Cy5 signals, was used.

**smFRET data analysis**. Single molecule fluorescence traces were extracted from the movies, and those showing six photobleaching steps of Cy3 were selected in case of equilibrium or flow-in PCNA loading measurements. In case of PCNA unloading measurements, as the photobleaching steps could not be observed, traces showing fluorescence intensity comparable to that of six Cy3 dyes under the same excitation condition were selected. Typically, unloading occurred much faster than photobleaching under our excitation condition (Supplementary Fig. 4), preventing false assignment of photobleaching events as unloading events. From the collected traces, we constructed FRET histograms and heat maps of FRET population dynamics, where the traces were synchronized by automatic detection of the PCNA association or dissociation moments, as described earlier[46]. We performed hidden Markov modeling using a program with empirical Bayesian inference developed by Gonzalez group (http://ebfret.github.io/)[47].

## Data availability

Data supporting the findings of this manuscript are available from the corresponding authors upon reasonable request. A reporting summary for this Article is available as a Supplementary Information file. The source data underlying Figs. 1b-d, 1i-n, 2f-g, 2b-g, 3b-e, 5b-f, 6a-h and Supplementary Figs. 1a-g, 2a, 2d-j, 3b-c, 3e-f, 4a, 5a-f, 6a-i are provided as a Source Data file.

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

## Acknowledgements

We thank members in the Center for Genomic Integrity, IBS for helpful discussions and comments on the manuscript. We also thank for Stephen P. Bell for providing us expression plasmids of yeast PCNA and TALE protein. K. Myung especially thanks E. Cho. Eunjin Ryu was supported by Global PhD Fellowship (NRF-2017H1A2A1044961). H.K. was supported by National Research Foundation of Korea (2017R1D1A1B03036239, 2017M3A9E2062181 and 2018R1A5A1024340). This research was mainly supported by the Institute for Basic Science (IBS-R022-D1). This work was also partially supported by UNIST research fund (1.180063).

## Author contribution

M.-S.K. performed most of cell-based experiments. E.R. purified most of human proteins and performed most of in vitro experiments. S.-W.L. performed single molecule experiments. J.P. performed unloading experiments of poly-ub PCNA. S.K. performed in vitro experiments with purified yeast proteins. N.Y.H. provided DNA constructs for this study and performed several cell-based experiments. J.S.R. performed flow cytometric analysis. Y.J. K. and J.K. provided reagents and proteins for several in vitro experiments. M.A-R. identified E1173K mutation of ATAD5 in human uveal (intraocular) melanoma. S.H.P. and K.-Y.L. made human cell lines for an EdU incorporation experiment. M.-S.K., E.R., S.-W.L., J. P., H.K., and S.K. designed the experiments and analyzed the data. S.K. and H.K. composed the manuscript with input from all authors. S.K., H.K., and K.M. directed the project.

## Additional information

**Competing interests:** The authors declare no competing interests.

