## [Peer Review File · Nature Communications]

Reviewers' Comments:

Reviewer #1:

Remarks to the Author:

This manuscript by Kang and colleagues addresses an important and still unresolved question, namely which RFC-like (RLC) complex unloads PCNA clamp and what is the unloading mechanism. Previous reports from the authors' lab suggested the importance of timely disassembly of the replication machinery and pointed to a role played by ATAD5 in this process. Here, the authors combined cellular, biochemical and single-molecule analysis to establish that (1) ATAD5-RLC has a unique PCNA unloading activity in cells and is a biochemically competent PCNA unloader, (2) that PCNA-unloading activity of ATAD5 can be functionally separated from its de-ubiquitination function, and (3) mapped the ATAD5 regions important for function. The description of the RFC 1 and ATAD5 truncated constructs that yield large quantities of stable and active proteins is an additional value added aspect of this work.

For most part, the experiments are well designed and expertly executed. Combination of the cellular and in vitro studies is a definite strength of this paper. And the model proposed by the authors is very convincing. The work will provide an important new information to the field and will be of significant interest to a broad readership of Nature Communications.

My only concern is with the description of the single-molecule FRET studies. The authors need to clarify how they calculated FRET for 6 FRET donors (Cy3s on PCNA) per one FRET acceptor (Cy5 on DNA) arrangement and whether they had to modify the trajectories to make ebFRET program to accept them. It is hard to imagine that the data shown in Fig 4 are indeed the raw, unmodified data with 6 Cy3 dyes – without removing the signal from 5 Cy3 dyes that do not transfer energy, this reviewer cannot see how the FRET of 0.6 can be achieved. The two papers the authors cite (Kim et al 2012 and 2014) used an experimental setup with a single donor and single acceptor of FRET. There have been reports where multiple acceptors were used with one donor, and the FRET calculations in such systems are similar to one donor per one acceptor calculation. IMO, more steps have to be taken to calculate the FRET with 6 donors per one acceptor. Please discuss the details.

Minor points:

From the single-molecule experiments, the Cy3-labeled PCNA seems functional. It is necessary, however, to show that it is as functional as the unlabeled protein.

The data fitting in Fig. 4m look a bit odd. I would not place much trust in the numeric outcome of the fitted curve. Since the parameters stemming from the data fitting are not used quantitatively, it may be prudent to remove the fit. Alternatively, a more appropriate model needs to be selected.

Reviewer #2:

Remarks to the Author:

PCNA functions as a platform for many proteins involved in DNA metabolism of replicating chromosomal DNA. Thus, its proper loading and unloading during the cell cycle is crucial to maintain integrity of the chromosome. Dr. Myung's group has demonstrated the PCNA unloading activity using purified ATAD5-RLC (Elg1-RLC in yeast), one of the RFC-like complexes (RLCs) by several biochemical approaches. The points of their observations were: 1) ATAD5-RLC (Elg1-RLC) is the only PCNA unloader among RLCs in vitro. 2) Structures of DNA substrates would not affect the unloading activity. 3) The ATPase motif ATAD5 and its association with RFC small subunits are crucial for the activity. 4) ATAD5-RLC (Elg1-RLC) could unload mono-, poly-ubiquitinated or SUMOylated PCNA, which occur upon DNA damage. 5) ATAD5-RLC unloads PCNA without ATP hydrolysis step as observed by single molecule imaging. 6) Some yeast replication proteins, for

example, Fen1 and Ligase1 could inhibit PCNA unloading by Elg1-RLC, suggesting their regulatory roles for the unloading. This study will be the first comprehensive analysis of PCNA unloading with purified proteins and potentially suitable for *Nat. Commun.*, if compelling data were provided here. However, although a significant amount of work has been presented in this manuscript, all of the demonstrated data are single representative figures without any quantitative and statistical information, except for the single molecule analysis in Fig. 4. The authors have discussed the levels of retained PCNA on DNA from single experiments with separated immunoblotting results frequently without quantitation standards, for example, in Fig. 1e, Fig. 3d, Fig. 5f, Fig. 6d, e, Fig. S1b and c, and Fig. S1g. Even from the same immunoblotting, it is hard to distinguish the differences of proteins levels just by blotting images, for example, all siATAD5 experiments (Fig. 2e, f, Fig. 3b, Fig. S2f, Fig. S5d), modified PCNA experiments (Fig. 5c, d, f, Fig. S5b, and Fig. S6h), and yeast Elg1-RLC experiments (Fig. 6c, e). To evaluate significance and reliability of those data, quantitated amounts of loaded PCNA in each experiment with standard PCNA of a known amount and their statistical processing should be necessary. In addition, I have a number of questions to be explained in this manuscript. From these incompleteness of this manuscript, I could not recommend it to *Nat. Commun.* The problems are summarized as major points below.

Major point 1

As I mentioned, quantitative data are mostly absent from this paper. I further point out a problem in the experiment of Fig. S3c and d for the S phase population by depletion of ATAD5 and ectopic expression of its ATPase-defective mutant, in which the differences were not significant enough to lead the authors arguments. Their multiple measurements and statistical information are necessary. In addition, expression levels of ATAD5 proteins in these experiments should be indicated.

Major point 2

The authors used the N-terminal truncations of ATAD5 and RFC1 for most of the experiments to make their purification easier. Since their 40-50% regions from the N-terminals are absent, I am afraid that some of their crucial functions may be affected by these truncations, though the authors mentioned that apparent properties of the truncated versions were similar as the full-length ones. However, the DNA ligase-like DNA binding motif of RFC1 (MCB, 1995, 15, 4661) and the SUMO-interacting motifs (SIMs) of ATAD5 (NAR, 2017, 45, 3189) are missing. I request the authors to repeat the crucial experiments with the full-length proteins, for example, Fig. 1e, Fig. 3e, Fig. 4, and Fig. 5c, d and f.

Major point 3

Previous literatures demonstrated PCNA unloading with purified human RFC (PNAS, 1996, 93, 12896; Genes Cells, 1996, 1, 101, eLife 2: e00278) and with purified yeast Ctf18-RLC in presence of RPA (MCB, 2005, 25, 5445). However, the authors showed that PCNA was not unloaded at all by RFC (Fig. 1e) or by Ctf18-RLC even in the presence of RPA (Fig. S6b). This is one of the crucial observation in this paper that ATAD5 is the only PCNA unloader (Page 8, line 5), and that only Elg1-RLC has PCNA-unloading activity among yeast clamp loader complexes (Page 16, the last sentence). However, this argument ignores discrepancies between this work and previously published works. These problems should be explained reasonably.

Major point 4

From the notion that PCNA unloading should be prevented during DNA synthesis DNA to prevent futile PCNA-loading and unloading cycle (Page 8, lines6-7), the authors studied the unloading with DNA substrates harboring a nick or a 11-nucleotide gap, and observed efficient unloading from both DNA. Then, they concluded that DNA structures, on which PCNA remains attached, did not significantly affect PCNA-unloading by ATAD5-RLC (Page 16, lines 7-8). However, these structures will not represent completion of DNA synthesis. Indeed, it has been reported that PCNA unloading by Elg1-RLC required the ligation of Okazaki fragment (Cell rep, 2015, 12, 774). Thus, analysis of PCNA-unloading from DNA, in which the nick has been ligated after PCNA loading, is necessary to test the structural differences of replicating and replicated DNA.

Major point 5

The authors found that Fen1 could inhibit the PCNA unloading and suggested that Fen1 inhibits PCNA unloading during DNA replication (Page 18, lines 2-3). The similar mechanism by DNA

polymerase δ for PCNA unloading by RFC has been published by Hedglin et al. (eLife 2: e00278). Thus, competition of PCNA binding by several PCNA interacting proteins and loader complexes will be a general mechanism to regulate PCNA cycling, not only for ATAD4-RLC. This point should be considered to revise this discussion.

In addition, I have several minor comments to be considered.

Minor point 1

In Fig. 1b, Fig. 2e, f, Fig. 3b, the leftmost lanes are the controls without ATAD5 depletion. Surprisingly, no chromatin-bound PCNA was detected in these untreated 293T cells, though a significant population of the cells should be in S phase, in which PCNA should exist on the replicating chromatin. Indeed, chromatin-bound PCNA was detected in the controls in Fig. S2f and Fig. S5d. These inconsistencies should be explained from a technical view.

Minor point 2

PCNA unloading by yeast Elg1-RLC seems to be less efficient than by ATAD4-RLC, as the used concentrations were 75-150nM for the former and 0.15-0.6nM for the latter. This point should be explained clearly. I am afraid that this might occur by used of truncated version of ATAD5-RLC, in which some negative regulatory domains would be removed from the intact one.

Minor point 3

Images of H3 in Fig. 1b were too dark to evaluate the amounts of the chromatin fraction. Use an appropriate exposure for the data.

Minor point 4

Numbers of primers annealed with 5'-biotinylated single stranded DNA substrate were described as "ten" at Page 7 line16 and "nine" at page 29 line 16. Indicate a precise number there. It is also necessary to describe their sequences in the supplementary information.

Minor point 5

At Page 9, lines 19-20, "ATAD5 (693-1719) bound to a lesser amount of RFC2-5 compared to ATAD5 (693-1844)" and Page 17, lines 9-10 "Elg1 (Δ C756)-RLC showed a sub-stoichiometric amount of Rfc2-5". I could not agree with these conclusions only from the images of Fig. S2c and Fig. S6f. Indicate more apparent data.

Minor point 6

At Page 16, lines 10-11, "To examine this possibility, we took advantage of the in vitro DNA replication system of budding yeast" will be "To examine this possibility, we took advantage of budding yeast replication proteins"

Minor point 7

At Page 18, lines 1-2, "By contrast, a mutation in the nuclease motif of Fen1 (CAT) did not inhibit PCNA unloading" should be "By contrast, a mutation in the nuclease motif of Fen1 (CAT) did inhibit PCNA unloading".

Minor point 8

"N693" in Fig. S5d has not been described in this text. I guess this will be the ATAD5 fragment from 1 to 693 aa with some protein tag, but it should be specified.

Reviewer #3:

Remarks to the Author:

Review of manuscript: "Regulation of PCNA cycling on replicating DNA by RFC and RFC-like complexes" by Mi-Sun Kang et al.

In this manuscript the authors take a biochemical approach to investigate the relative roles of RFC and RFC-like complexes (RLCs) in PCNA metabolism. They show that whereas RFC and the CTF18 RLC load PCNA, they do not unload it from chromatin. The Rad24 RLC loads the 9-1-1 complex, and the ATAD5-Elg1 RLC is the only one that unloads PCNA from the DNA. By carrying out single-molecule measurements, they arrive at the conclusion that loading by RFC can be divided into three steps, whereas unloading by ATAD5 shows only two. Unloading was independent of the DNA structure, and was not affected by PCNA modifications.

In general, the analysis of these complexes is interesting and important and the data presented

are sound. The paper will be of interest to a wide readership. I have only a few queries, and many minor corrections, mainly of the English.

Major:

1) None of the figures includes quantitation. This is unacceptable, particularly for experiments such as PCNA loading/unloading in which the authors derive conclusions based on quantities. Some of the "control proteins" that could be used as reference for quantitation are extremely overexposed, and cannot serve as a measure for equal loading.

2) The role played by ATP in the activity of the ATAD5-Elg1-RLC is confusing to this referee. The authors conclude, on one hand, that the WalkerA and WalkerB motifs are essential for PCNA unloading activity (the mutations accumulate PCNA at the same level as cells depleted of ATAD5). Moreover, defects in the ATPase cause reduced EdU incorporation in vivo in dividing cells (Figure 3). On the other hand, the authors show that ATP-Gamma-S only reduces activity and can be easily overcome by additional ATAD5 complex (Figure 3e) and FRET experiments show that ATP-Gamma-S did not affect the high-FRET state. The authors then say that ATP hydrolysis may not be required to bring down PCNA, only to dissociate ATAD5 from it. But how do they then explain the results obtained by the Walker A/B mutants? If ATP hydrolysis was only a post-unloading step, the mutants should still be able to unload PCNA from chromatin.

3) The effects of CM sequences on the interaction with the small Rfc subunits and on PCNA unloading are not internally consistent, and remain unexplained.

4) To study the possible role of ATAD5 independently of the small Rfc subunits, instead of mutating the interphase, the authors can attempt unloading assays in the absence of the small subunits.

Minor:

4) What is the nature of the relatively strong bands of about 130 MDa in Figure 1c, or the asterisk-marked band in Suppl. Figure 2C?

5) The model represented by Suppl. Figure 3F cannot be understood from the drawing nor from the legend.

6) There is a need for English editing throughout the paper. Here are some examples:

Page 4, line 17: "experiments"

Page 4, line 18: "experiments"

Page 4, last line: "mammalian cells are sensitive"

Legend of Figure 1: "is conferred by the C-terminal domain"

Page 7, line 12: "we purified the N-terminal-deleted..."

Page 9, line 16: delete "were"

Page 10, line 12: form

Page 10, 4 lines before end: quote Supplementary Figure 3.

Page 17, line 7: something is wrong with the sentence: "those Walker A and Walker B..."

Reviewer #4:

Remarks to the Author:

The manuscript by Kang et al. aims to dissect how distinct clamp loaders/unloaders mediate the lifetime of PCNA on DNA. The manuscript encompasses a very large body of biochemical work on the ATAD5-RLC. Detailed studies addressing the critical questions in this field have remain elusive, largely due to the difficulty in obtaining ATAD5-RLC for biochemical experiments. The authors have made a significant leap in purifying a truncated form of human ATAD5-RLC for biochemical experiments and some of the work presented characterizes how the ATAD5 subunit is incorporated into a functional RLC complex with unloading activity. However, at this stage, the major findings of the manuscript are limited in their novelty and scope (see details below) and, hence, do not warrant publication in Nature Communications in the current format.

Major Concerns:

1. Based largely on the results presented in Figure 1E, the authors argue that ATAD5-RLC is the only PCNA unloader. This contradicts a large body of work from many labs (O'Donnell, Hurwitz,

Benkovic, Fanning, etc) which independently demonstrated that, in addition to loading PCNA onto DNA, human RFC also unloads PCNA from DNA and such activity is robust. This work was neither cited nor discussed in the manuscript. The discrepancy between the current manuscript and the previous work may lie in the experimental design.

The way the substrate is depicted in Figure 1D, there is nothing to prevent PCNA from sliding off the 3' end of the 5'-biotinylated DNA substrate due to the absence of physical blocks (RPA, secondary DNA structures such as hairpins, etc.). Work from the van Oijen and Walter labs demonstrated that human PCNA rapidly diffuses along kilobases of DNA in a random manner (JBC 2009). After a 30 minute incubation at 37°C on a 1.4 kb DNA substrate, I would envision that a significant portion of the PCNA-bound DNA would lose its PCNA during the incubation simply by random diffusion (i.e., in a manner independent of any unloading activity). This behavior was not observed, suggesting that the DNA substrate may contain prominent intramolecular secondary structures. RFC-catalyzed unloading of PCNA is substrate-dependent. Furthermore, previous work from the Benkovic lab revealed that human RFC-catalyzed unloading of human PCNA from DNA is inhibited at ATP concentrations greater than 1 mM. For instance, increasing ATP concentration from 1 mM to 5 mM ATP inhibits RFC-catalyzed unloading ~4-fold (eLife 2013). The unloading experiments depicted in Figure 1E were carried out at 6 mM ATP (well above physiological levels) where RFC-catalyzed unloading of PCNA is significantly inhibited. Thus, RFC unloading activity is inhibited under the experimental conditions and possibly by the DNA substrate itself. As is, any conclusion/suggestion comparing the relative PCNA unloading activities of RFC and ATAD5 are restricted to the experiment described in Figure 1D – E for the 1.4 kb DNA substrate and are not generally applicable.

2. The smFRET results described in Figure 4 pertaining to RFC-catalyzed loading of PCNA confirm previous findings from ensemble FRET measurements on human RFC by the Benkovic lab (eLife, 2013, Biochemistry 2017). Namely, human PCNA loading occurs through two intermediate states on DNA, separated by ATP hydrolysis and, hence, human RFC-catalyzed loading of human PCNA stalls at the first intermediate state in the presence of ATP γ S.

3. The results described in Figure 5 and 6 pertaining to unloading of PCNA by ATAD5/ELG1-RLC were previously observed for unloading of human PCNA by human RFC; work from the Benkovic lab demonstrated that the robust PCNA unloading activity of human RFC is independent of PCNA monoubiquitination (PNAS 2016) and inhibited by DNA replication proteins that interact with PCNA through PIP boxes (eLife 2013). Hence, the only difference between unloading of PCNA by RFC and ATAD5-RLC is that RFC-catalyzed unloading of PCNA is dependent on ATP hydrolysis whereas ATP hydrolysis is not required for ATAD5-RLC catalyzed unloading of PCNA, as indicated in the current manuscript.

The comparative unloading experiments should be repeated at ATP concentrations < 1 mM on a minimal DNA substrate (similar to the one described in Supplementary Figure 1F) where both RFC and ATAD5-RLC catalyzed unloading of PCNA are observed. The unloading activities should be quantitatively assessed. This DNA substrate should then be systematically altered (i.e., increasing/decreasing the primer lengths, etc.) and the relative unloading activities of RFC and ATAD5-RLC quantitatively compared to decipher where the differences in unloading activities, if any, lie.

Minor Concerns

1. The authors claim that PCNA unloading requires the ATAD5-RLC complex, not just ATAD5 alone. This conclusion cannot be reached without an experiment where PCNA unloading is monitored in the presence of only the wild-type ATAD5 subunit.

2. Functionality of the Cy3-labeled PCNA. The ability of PCNA to interact with other proteins, be loaded onto DNA, etc. is sensitive to amino acid mutations and modifications. To my knowledge, this is the first report where the native, exposed cysteines of PCNA were labeled with Cy3. It is customary to demonstrate that such labeling does not affect the ability of PCNA to function

properly. Also, without correcting for the absorbance of Cy3 at 280 nm or determining the concentration of the protein via bradford, the percent labeling is likely an underestimate.

The major points of revision are the followings.

First, as reviewers #2 and #3 pointed out, our initial manuscript lacked quantification of the data. To strengthen our arguments, we included quantification results with statistical analysis in the revised manuscript.

Second, as reviewer #2 and #4 pointed out, there was a discrepancy between our results and previous reports regarding PCNA unloading activity of RFC. Although we did not observe RFC-mediated PCNA unloading, there are earlier studies reporting PCNA-unloading activity of RFC. To address this discrepancy, we more carefully examined RFC-mediated PCNA unloading in vitro (Supplementary Fig. 1k-n and Supplementary Fig. 4c). We found that RFC indeed has PCNA unloading activity, but the activity is significantly lower than that of ATAD5-RLC (Supplementary Fig. 1k, l and n). RFC unloads PCNA more efficiently from nicked DNA compare to gapped DNA, but the unloading efficiency is still much lower than that of ATAD5-RLC (Supplementary Fig. 1m, n, Supplementary Fig. 2j). Because of such characteristics of RFC for PCNA unloading, we had not observed significant PCNA-unloading activity of RFC in our initial manuscript. We revised our manuscript based on those new findings. Although RFC has little PCNA-unloading activity, our argument is still valid to claim that ATAD5-RLC possesses the most potent PCNA-unloading activity among clamp-loader complexes.

Third, as reviewer #2 pointed out, we used N-terminal deleted ATAD5 for in vitro studies, because our *in vitro* and cell-based experiments showed that ATAD5 (Δ N692)-RLC is sufficient for PCNA unloading. As the reviewer requested, we purified full-length ATAD5-RLC and performed key experiments (Supplementary Fig. 2i-j, Supplementary Fig. 3f and Supplementary Fig. 5c and e). We found that both full-length ATAD5-RLC and ATAD5 (Δ N692)-RLC unload PCNA or Ub-PCNA. We believe that these results support our argument that N-terminal domain of ATAD5-RLC is not essential for its PCNA-unloading activity.

In addition to above major points, we have addressed each comment of the reviewers in detail. We also revised “Figure Legends” and “Methods” and fixed miscellaneous errors in the manuscript.

Here are our responses to each comment of the reviewers.

Reviewer #1.

Reviewer #1 (Remarks to the Author):

This manuscript by Kang and colleagues addresses an important and still unresolved question, namely which RFC-like (RLC) complex unloads PCNA clamp and what is the unloading mechanism. Previous reports from the authors’ lab suggested the importance of timely disassembly of the replication machinery and pointed to a role played by ATAD5 in this process. Here, the authors combined cellular, biochemical and single-molecule analysis to establish that (1) ATAD5-RLC has a unique PCNA unloading activity in cells and is a biochemically competent PCNA unloader, (2) that PCNA-unloading activity of ATAD5 can be functionally separated from its de-ubiquitination function, and (3) mapped the ATAD5 regions important for function. The description of the RFC1 and ATAD5 truncated constructs that yield large quantities of stable and active proteins is addition value added aspect of this work. For most part, the experiments are well designed and expertly executed. Combination of the cellular and in vitro studies is a definite strength of this paper. And the model proposed by the authors is very convincing. The work will provide an important new information to the field and will be of significant interest to a broad readership of Nature Communications.

My only concern is with the description of the single-molecule FRET studies. The authors need to clarify how they calculated FRET for 6 FRET donors (Cy3s on PCNA) per one FRET acceptor (Cy5 on DNA) arrangement and whether they had to modify the trajectories to make ebFRET program to accept them. It is hard to imagine that the data shown in Fig 4 are indeed the raw, unmodified data with 6 Cy3 dyes – without removing the signal from 5 Cy3 dyes that do not transfer energy, this reviewer cannot see how the FRET of 0.6 can be achieved. The two papers the authors cite (Kim et al 2012 and 2014) used an experimental setup with a single donor and single acceptor of FRET. There have been reports where multiple acceptors were used with one donor, and the FRET calculations in such systems are similar to one donor per one acceptor calculation. IMO, more steps have to be taken to calculate the FRET with 6 donors per one acceptor. Please discuss the details.

Minor points:

From the single-molecule experiments, the Cy3-labeled PCNA seems functional. It is necessary, however, to show that it is as functional as the unlabeled protein.

The data fitting in Fig. 4m look a bit odd. I would not place much trust in the numeric outcome of the fitted curve. Since the parameters stemming from the data fitting are not used quantitatively, it may be prudent to remove the fit. Alternatively, a more appropriate model needs to be selected.

Responses to Major Concern)

We understand the concern with possible effects of multiple fluorophores presented in our single PCNA complex. When multiple FRET acceptors are present at equal distance from one donor, the apparent FRET efficiency does not convert to the distance given by the equation for FRET efficiency with common Förster radius because the probability of energy transfer depends on (proportional to) the number of acceptors, resulting in higher apparent FRET efficiency than when a single pair of dyes are at the same distance. Now, if we imagine multiple donors at equal distance from one acceptor, the acceptor will fluoresce proportional to the frequency of energy transfer events from all donors. Under typical excitation conditions, the excited lifetime of the donor (~ nanoseconds) is much shorter than the waiting time between excitation events (~ sub-millisecond) and thus only one donor is likely to be excited at each moment. For the acceptor, it will appear as if a single donor is hopping around in space just with more frequent excitation events, thus giving the same FRET efficiency as that of a single pair of dyes at the same distance.

One effect to consider is the homoFRET between donor dyes. homoFRET leads to the change in fluorescence polarization but not the total intensity. Thus, we expect our multiple-donor labeling scheme to result in the average FRET efficiency for all donor-acceptor combinations.

We experimentally tested how the number of donor dyes affects the apparent FRET efficiency by constructing FRET evolution maps from different groups of traces containing 1, 2, or 3 Cy3 molecules, which were distinguished by the number of observed photobleaching steps. As presented in the above figure, the FRET evolution map from traces with fewer donor dyes shows broader pattern of FRET population density but the observed FRET levels remain consistent with the high FRET level (0.62) of the loading intermediate 2 and the low FRET level (0.34) of the loaded state, respectively, observed from the FRET evolution map with larger number of donor dyes. The broadening of FRET levels stems from the lower S/N and the variant position of Cy3 dyes on the PCNA ring. The FRET density maps from traces with 2 and 3 Cy3 dyes better reveal the short-lived initial mid-FRET population, similar to the map in Figure 4. Thus, we conclude that our labeling scheme using multiple Cy3 dyes reveals more precise FRET dynamics representing the average distance between the PCNA ring and the label on the DNA.

Above is an example FRET trace showing the loading of a PCNA trimer that contains 6 Cy3 dyes, represented by 6 photobleaching steps. We included these results in the revised manuscript as Supplementary Figure 4b, c and referred to this in the text as below.

Page 13, line 16-20

“Thus, we fully labeled all three subunits with Cy3 at the two exposed cysteine residues (Cys62, Cys81), which allows for measuring the average distance between the PCNA ring and the label on the DNA.”

→ “Thus, we fully labeled all three subunits with Cy3 at the two exposed cysteine residues (C62, C81) (Supplementary Fig. 4a), which allows for measuring the average distance between the PCNA ring and the label on the DNA without being obscured by the varying position of the label (Supplementary Fig. 4b-c).”

Minor point one

From the single-molecule experiments, the Cy3-labeled PCNA seems functional. It is necessary, however, to show that it is as functional as the unlabeled protein.

For this concern, we now included our results comparing the loading capacity of unlabeled PCNA and labeled PCNA (Supplementary Fig. 4a). Cy3-labeled PCNA was loaded in less but substantial amount, possibly due to the presence of dyes affecting the binding affinity to RFC complex. Labeling does not compromise the functionality of PCNA as both loading and unloading activities were confirmed from single molecule observations in consistent manner with ensemble level data.

Minor point two

The data fitting in Fig. 4m look a bit odd. I would not place much trust in the numeric outcome of the fitted curve. Since the parameters stemming from the data fitting are not

used quantitatively, it may be prudent to remove the fit. Alternatively, a more appropriate model needs to be selected.

We agree that the histograms in Fig. 4j and 4m do not fit well enough to gamma distribution. It is possible that there are multiple conformational pathways through which the ATAD5-RLC-bound open PCNA ring dissociates from the DNA junction, resulting in mixed statistical behaviors. Thus, we removed the gamma distribution fitting in the revised manuscript. However, we still appreciate the clear distinction between the dwell times of LI1 and LI2 states (Fig. 4h and 4i) and the dwell time of UI state (Fig. 4j and 4m); the latter shows a distribution peaked at non-zero value while the former shows an exponential decay. We interpret this difference as the existence of multiple steps in the course of PCNA dissociation from DNA. The coincident peak positions in Fig. 4j and 4m suggest that the PCNA dissociation dynamics is not coupled to the hydrolysis of ATP. Accordingly, we revised the text as below.

Page 15, line 12-14

“By contrast, the dwell time distribution of UI was far from being exponential, and followed more of a gamma distribution (Fig. 4j), implying the presence of multiple catalytic steps within UI during the unloading process.”

→ “By contrast, the dwell time distribution of UI was far from being exponential and showed a peak at non-zero time (Fig. 4j), which suggest the presence of multiple catalytic steps within UI during the unloading process.”

Page 15, line 26 ~ Page 16, line 2

“Furthermore, the dwell time distribution of UI was similar between the ATP reaction and the ATP- γ -S reaction (Fig. 4j and m).”

→ “Furthermore, the dwell time distribution of UI showed a peak at similar time between the ATP and the ATP- γ -S reactions (Fig. 4j and m).”

Legend of Figure 4. h-j

Following sentence was deleted.

“UI was fitted to gamma distribution, $Bt^{-1}e^{-t}$, and shape parameter α and inverse scale parameter β are shown.”

Reviewer #2.

PCNA functions as a platform for many proteins involved in DNA metabolism of replicating chromosomal DNA. Thus, its proper loading and unloading during the cell cycle is crucial to maintain integrity of the chromosome. Dr. Myung's group has demonstrated the PCNA unloading activity using purified ATAD5-RLC (Elg1-RLC in yeast), one of the RFC-like complexes (RLCs) by several biochemical approaches. The points of their observations were; 1) ATAD5-RLC (Elg1-RLC) is the only PCNA unloader among RLCs in vitro. 2) Structures of DNA substrates would not affect the unloading activity. 3) The ATPase motif ATAD5 and its association with RFC small subunits are crucial for the activity. 4) ATAD5-RLC (Elg1-RLC) could unload mono-, poly-ubiquitinated or SUMOylated PCNA, which occur upon DNA damage. 5) ATAD5-RLC unloads PCNA without ATP hydrolysis step as observed by single molecule imaging. 6) Some yeast replication proteins, for example, Fen1 and Ligase1 could inhibit PCNA unloading by Elg1-RLC, suggesting their regulatory roles for the unloading. This study will be the first comprehensive analysis of PCNA unloading with purified proteins and potentially suitable for Nat. Commun, if compelling data were provided here. However, although a significant amount of work has been presented in this manuscript, all of the demonstrated data are single representative figures without any quantitative and statistical information, except for the single molecule analysis in Fig. 4. The authors have discussed the levels of retained PCNA on DNA from single experiments with separated immunoblotting results frequently without quantitation standards, for example, in Fig. 1e, Fig. 3d, Fig. 5f, Fig. 6d, e, Fig. S1b and c, and Fig. S1g. Even from the same immunoblotting, it is hard to distinguish the differences of proteins levels just by blotting images, for example, all siATAD5 experiments (Fig. 2e, f, Fig. 3b, Fig. S2f, Fig. S5d), modified PCNA experiments (Fig. 5c, d, f, Fig. S5b, and Fig. S6h), and yeast Elg1-RLC experiments (Fig. 6c, e). To evaluate significance and reliability of those data, quantitated amounts of loaded PCNA in each experiment with standard PCNA of a known amount and their statistical processing should be necessary. In addition, I have a number of questions to be explained in this manuscript. From these incompleteness of this manuscript, I could not recommend it to Nat. Commun. The problems are summarized as major points below.

Major point 1

As I mentioned, quantitative data are mostly absent from this paper. I further point out a problem in the experiment of Fig. S3c and d for the S phase population by

depletion of ATAD5 and ectopic expression of its ATPase-defective mutant, in which the differences were not significant enough to lead the authors arguments. Their multiple measurements and statistical information are necessary. In addition, expression levels of ATAD5 proteins in these experiments should be indicated.

Major point 2

The authors used the N-terminal truncations of ATAD5 and RFC1 for most of the experiments to make their purification easier. Since their 40-50% regions from the N-terminals are absent, I am afraid that some of their crucial functions may be affected by these truncations, though the authors mentioned that apparent properties of the truncated versions were similar as the full-length ones. However, the DNA ligase-like DNA binding motif of RFC1 (MCB, 1995, 15, 4661) and the SUMO-interacting motifs (SIMs) of ATAD5 (NAR, 2017, 45, 3189) are missing. I request the authors to repeat the crucial experiments with the full-length proteins, for example, Fig. 1e, Fig. 3e, Fig. 4, and Fig. 5c, d and f.

Major point 3

Previous literatures demonstrated PCNA unloading with purified human RFC (PNAS, 1996, 93, 12896; Genes Cells, 1996, 1, 101, eLife 2: e00278) and with purified yeast Ctf18-RLC in presence of RPA (MCB, 2005, 25, 5445). However, the authors showed that PCNA was not unloaded at all by RFC (Fig. 1e) or by Ctf18-RLC even in the presence of RPA (Fig. S6b). This is one of the crucial observation in this paper that ATAD5 is the only PCNA unloader (Page 8, line 5), and that only Elg1-RLC has PCNA-unloading activity among yeast clamp loader complexes (Page 16, the last sentence). However, this argument ignores discrepancies between this work and previously published works. These problems should be explained reasonably.

Major point 4

From the notion that PCNA unloading should be prevented during DNA synthesis DNA to prevent futile PCNA-loading and unloading cycle (Page 8, lines6-7), the authors studied the unloading with DNA substrates harboring a nick or a 11-nucleotide gap, and observed efficient unloading from both DNA. Then, they concluded that DNA structures, on which PCNA remains attached, did not significantly affect PCNA-unloading by ATAD5-RLC (Page 16, lines 7-8). However, these structures will not represent completion of DNA synthesis. Indeed, it has been reported that PCNA unloading by Elg1-RLC required the ligation of Okazaki fragment (Cell rep, 2015, 12, 774). Thus, analysis of PCNA-unloading from DNA, in which the nick has been ligated after PCNA loading, is necessary to test the structural differences of replicating and replicated DNA.

Major point 5

The authors found that Fen1 could inhibit the PCNA unloading and suggested that Fen1 inhibits PCNA unloading during DNA replication (Page 18, lines 2-3). The similar mechanism by DNA polymerase δ for PCNA unloading by RFC has been published by Hedglin et al. (eLife 2: e00278). Thus, competition of PCNA binding by several PCNA interacting proteins and loader complexes will be a general mechanism to regulate PCNA cycling, not only for ATAD4-RLC. This point should be considered to revise this discussion.

In addition, I have several minor comments to be considered.

Minor point 1

In Fig. 1b, Fig. 2e, f, Fig. 3b, the leftmost lanes are the controls without ATAD5 depletion. Surprisingly, no chromatin-bound PCNA was detected in these untreated 293T cells, though a significant population of the cells should be in S phase, in which PCNA should exist on the replicating chromatin. Indeed, chromatin-bound PCNA was detected in the controls in Fig. S2f and Fig. S5d. These inconsistencies should be explained from a technical view.

Minor point 2

PCNA unloading by yeast Elg1-RLC seems to be less efficient than by ATAD4-RLC, as the used concentrations were 75-150nM for the former and 0.15-0.6nM for the latter. This point should be explained clearly. I am afraid that this might occur by used of truncated version of ATAD5-RLC, in which some negative regulatory domains would be removed from the intact one.

Minor point 3

Images of H3 in Fig. 1b were too dark to evaluate the amounts of the chromatin fraction. Use an appropriate exposure for the data.

Minor point 4

Numbers of primers annealed with 5'-biotinylated single stranded DNA substrate were described as "ten" at Page 7 line16 and "nine" at page 29 line 16. Indicate a precise number there. It is also necessary to describe their sequences in the supplementary information.

Minor point 5

At Page 9, lines 19-20, "ATAD5 (693-1719) bound to a lesser amount of RFC2-5 compared to ATAD5 (693-1844)" and Page 17, lines 9-10 "Elg1 (Δ C756)-RLC showed

a sub-stoichiometric amount of Rfc2-5". I could not agree with these conclusions only from the images of Fig. S2c and Fig. S6f. Indicate more apparent data.

Minor point 6

At Page 16, lines 10-11, "To examine this possibility, we took advantage of the in vitro DNA replication system of budding yeast" will be "To examine this possibility, we took advantage of budding yeast replication proteins"

Minor point 7

At Page 18, lines 1-2, "By contrast, a mutation in the nuclease motif of Fen1 (CAT) did not inhibit PCNA unloading" should be "By contrast, a mutation in the nuclease motif of Fen1 (CAT) did inhibit PCNA unloading".

Minor point 8

"N693" in Fig. S5d has not been described in this text. I guess this will be the ATAD5 fragment from 1 to 693 aa with some protein tag, but it should be specified.

General suggestion

However, although a significant amount of work has been presented in this manuscript, all of the demonstrated data are single representative figures without any quantitative and statistical information, except for the single molecule analysis in Fig. 4. The authors have discussed the levels of retained PCNA on DNA from single experiments with separated

immunoblotting results frequently without quantitation standards, for example, in Fig. 1e, Fig. 3d, Fig. 5f, Fig. 6d, e, Fig. S1b and c, and Fig. S1g. Even from the same immunoblotting, it is hard to distinguish the differences of proteins levels just by blotting images, for example, all siATAD5 experiments (Fig. 2e, f, Fig. 3b, Fig. S2f, Fig. S5d), modified PCNA experiments (Fig. 5c, d, f, Fig. S5b, and Fig. S6h), and yeast Elg1-RLC experiments (Fig. 6c, e). To evaluate significance and reliability of those data, quantitated amounts of loaded PCNA in each experiment with standard PCNA of a known amount and their statistical processing should be necessary.

We apologize for the lack of quantification results in our previous manuscript. To address reviewer's concern, we included quantification results with statistical analysis in the revised manuscript. We present quantification data for experiments analyzing chromatin-bound PCNA (Fig. 1c, 2e-f, 3b, Supplementary Fig. 2e, Supplementary Fig. 3 c, and Supplementary Fig. 5f) and most of our in vitro experiments (Fig 1f, 2g, 3d-e, 5c-d, 5f, 6a-c, 6e-g, Supplementary Fig. 1e, Supplementary Fig. 1i-n, Supplementary Fig. 2h, 2j, Supplementary Fig. 3f, Supplementary Fig 5c, 5e, and Supplementary Fig. 6h). Numbers were presented as relative values compared to control. We believe that these analyses strengthen our arguments.

Major point 1

As I mentioned, quantitative data are mostly absent from this paper. I further point out a problem in the experiment of Fig. S3c and d for the S phase population by depletion of ATAD5 and ectopic expression of its ATPase-defective mutant, in which the differences were not significant enough to lead the authors arguments. Their multiple measurements and statistical information are necessary. In addition, expression levels of ATAD5 proteins in these experiments should be indicated.

As described above, we included quantitative results in the revised manuscript. In the case of Supplementary Fig. 3c, we repeated the experiment using cell lines expressing full-length ATAD5 or its EK mutant. New data set is presented in the revised manuscript. As the reviewer suggested, we included data showing expression levels of ATAD5. Expression levels of wild type ATAD5 and EK mutant were similar. Although the amount of exogenous ATAD5 was less than endogenous ATAD5, expression of wild type ATAD5 in ATAD5 depleted cells reduced the chromatin-bound PCNA and increased EdU incorporation. However, EK mutant did not reduce chromatin-bound PCNA and failed to increase EdU incorporation in ATAD5 depleted cells. We believe that these new results further strengthen our argument that “ATAD5-RLC-dependent PCNA unloading is important for proper DNA replication.”

Major point 2

The authors used the N-terminal truncations of ATAD5 and RFC1 for most of the experiments to make their purification easier. Since their 40-50% regions from the N-terminals are absent, I am afraid that some of their crucial functions may be affected by these truncations, though the authors mentioned that apparent properties of the truncated versions were similar as the full-length ones. However, the DNA ligase-like DNA binding motif of RFC1 (MCB, 1995, 15, 4661) and the SUMO-interacting motifs (SIMs) of ATAD5 (NAR, 2017, 45, 3189) are missing. I request the authors to repeat the crucial experiments with the full-length proteins, for example, Fig. 1e, Fig. 3e, Fig. 4, and Fig. 5c, d and f.

From the beginning of this study, we have been trying to purify full-length ATAD5-RLC from various systems. However, it has been hard to purify full-length ATAD5-RLC, because full-length ATAD5 was not expressed well and very unstable. Therefore, we identified minimal region of ATAD5 and used ATAD5 (Δ N692) for PCNA-unloading assays in the earlier version of the manuscript. We agree that PCNA-unloading experiments should be performed

with full-length ATAD5-RLC to address concerns raised by the reviewer. Therefore, we tried different approaches to purify ATAD5-RLC. Recently, we were able to purify full-length ATAD5-RLC using the combination of yeast and bacterial expression system. ATAD5 and RFC2-5 were co-expressed in budding yeast. Yeast cell lysate was prepared and mixed with bacterial cell lysate expressing RFC2-5 to supplement small RFC subunits. ATAD5-RLC was purified from the mixed lysate using FLAG affinity resin followed by SP Sepharose (Supplementary Fig. 2i). We also purified full-length RFC and ATAD5 (Δ N692)-RLC using the same approach (Supplementary Fig. 1a and Supplementary Fig. 2g). PCNA-unloading activities of full-length RFC (Supplementary Fig. 1f) and full-length ATAD5-RLC (Supplementary Fig. 2j) were examined. Full-length RFC did not reduce the amount of DNA-loaded PCNA at concentrations where ATAD5-RLC unloaded PCNA. ATAD5-RLC and ATAD5 (Δ N692)-RLC showed similar PCNA-unloading activity. Furthermore, results from ATP- γ -S experiment (Supplementary Fig. 3f) and Ub-PCNA unloading experiment (Supplementary Fig. 5c) show that biochemical properties of ATAD5-RLC and ATAD5 (Δ N692)-RLC are very similar to each other. These results strengthen our arguments that N-terminal domain of ATAD5 is not essential for PCNA-unloading activity of ATAD5-RLC. As reviewer pointed out, N-terminal domain of ATAD5 contains various conserved motifs. In addition to PCNA unloading, ATAD5 participates in various cellular processes such as Ub-PCNA de-ubiquitination and homologous recombination. We think that those N-terminal motifs might be important for those functions of ATAD5.

According to above results, we revised our manuscript as below.

Page 6, line 15-16

“To investigate the biochemical properties of ATAD5-RLC in PCNA unloading, we tried to purify full-length ATAD5. However, full-length ATAD5 was not expressed well and seemed relatively unstable (data not shown). Therefore, we mapped a minimal region of ATAD5 that is sufficient for PCNA unloading.”

→ “To investigate the biochemical properties of ATAD5-RLC in PCNA unloading, we first mapped a minimal region of ATAD5 that is sufficient for PCNA unloading.”

Page 10, line 23 ~ Page 11, line 3

Inserted following sentences:

“To more clearly define functional contributions of each ATAD5 domain for PCNA unloading, we purified ATAD5 (Δ N692) alone and full-length ATAD5-RLC using the same system (Supplementary Fig. 2i) and examine their PCNA-unloading activity (Supplementary Fig.2j). ATAD5 (Δ N692) alone did not show PCNA-unloading activity. PCNA unloading activity of full-length ATAD5-RLC and ATAD5 (Δ N692)-RLC were similar. Therefore, N-terminal domain of ATAD5 is not essential for PCNA unloading and PCNA unloading requires ATAD5-RLC, not ATAD5 alone.”

Page 13, line 6-7

Underlined words were inserted into the original sentence.

“ATP- γ -S also interfered with the efficient PCNA unloading by ATAD5-RLC (Fig. 3e and Supplementary Fig. 3f).”

Page 16, line 18-22

“Remarkably, ATAD5-RLC efficiently unloaded mono-ubiquitinated PCNA with the same efficiency as unmodified PCNA (Fig. 5c, d, and Supplementary Fig. 5b). This result suggests that PCNA mono-ubiquitination did not affect PCNA unloading by ATAD5-RLC.5b~c.”

→ “Remarkably, ATAD5 (Δ N692)-RLC efficiently unloaded mono-ubiquitinated PCNA (Fig. 5c-d, and Supplementary Fig. 5b). Unloading efficiency of full-length ATAD5-RLC was not significantly affected by PCNA mono-ubiquitination (Supplementary Fig. 5c). These results suggest that PCNA mono-ubiquitination did not significantly affect PCNA unloading by ATAD5-RLC.”

Page 17, line 3-5

Underlined words were inserted into the original sentence.

“After poly-ubiquitination, ATAD5 (Δ N692)-RLC or ATAD5-RLC was added and the release of poly-ubiquitinated PCNA was monitored using anti-Ub-PCNA antibody (Fig. 5f and Supplementary Fig. 5e).”

Major point 3

Previous literatures demonstrated PCNA unloading with purified human RFC (PNAS, 1996, 93, 12896; Genes Cells, 1996, 1, 101, eLife 2: e00278) and with purified yeast Ctf18-RLC in presence of RPA (MCB, 2005, 25, 5445). However, the authors showed that PCNA was not unloaded at all by RFC (Fig. 1e) or by Ctf18-RLC even in the presence of RPA (Fig. S6b). This is one of the crucial observation in this paper that ATAD5 is the only PCNA unloader (Page 8, line 5), and that only Elg1-RLC has PCNA-unloading activity among yeast clamp loader complexes (Page 16, the last sentence). However, this argument ignores discrepancies between this work and previously published works. These problems should be explained reasonably.

As reviewer pointed out, previous reports showed that human RFC could unload PCNA. To understand discrepancy between our results and previous studies, we carefully examined various experimental conditions for RFC-mediated PCNA unloading (Supplementary Fig. 1k~n). We examined PCNA-unloading from short DNA instead of 1.4 Kbps DNA. We reduced ATP concentration in PCNA-unloading reaction, because a previous report showed that RFC-mediated PCNA unloading could be inhibited by high ATP concentration (Fig. 1e and Supplementary Fig. 1k). Same as our initial results in the previous manuscript, RFC did not show apparent detectable PCNA-unloading activity. However, excessive amount of RFC reduced the amount of DNA-loaded PCNA (Supplementary Fig. 1l). When we used nicked DNA, RFC showed PCNA-unloading activity at lower concentration compared to gapped DNA (Supplementary Fig. 1m). However, RFC was significantly less efficient for PCNA unloading compared to ATAD5-RLC (Supplementary Fig. 1n, compare Supplementary Fig.

1k and Supplementary Fig. 1I). These results suggested that although RFC could unload PCNA as previously reported, ATAD5-RLC is a more potent PCNA unloader compared to RFC. We believe that RFC might function as a secondary PCNA unloader during DNA replication.

In the case of budding yeast clamp-loader complexes, one report showed that Ctf18-RLC could unload PCNA *in vitro*. In their report, RFC did not unload PCNA same as our results. However, they did not observe PCNA-unloading activity of Elg1-RLC contrary to our data. We believe that our results match with *in vivo* results because *elg1* deletion results in the accumulation of PCNA on the chromatin. Although we did not observe PCNA unloading by Ctf18-RLC, we described the possibility that Ctf18-RLC might function as a secondary PCNA unloader.

According to new results, we revised our manuscript as below.

Page 2, line 7-8

“ATAD5-RLC uniquely possesses PCNA unloading activity while RFC, CTF18-RLC, and RAD17-RLC catalyze clamp loading only.”

→ “ATAD5-RLC possesses the most potent PCNA unloading activity while RFC, CTF18-RLC, and RAD17-RLC mainly catalyze clamp loading.”

Page 6, line 9

“ATAD5-RLC is a specific PCNA unloader”

→ “ATAD5-RLC is a potent PCNA unloader”

Page 7, line 14-18

“We prepared 5'-biotinylated single-stranded DNA substrate (1.4 Kbs) on which ten 20-mer oligonucleotides were annealed with ~150-nucleotide gaps between them. This substrate DNA mimics primer-template junctions. The substrate DNA was then attached to the streptavidin-coated magnetic beads.”

→ “We prepared 130-mer DNA that contains 10 nucleotides gap. One end of the 130-mer DNA was biotinylated and attached to streptavidin-coated magnetic beads. Because the substrate DNA is short, we added a TALE-binding sequence to the opposite side of biotinylation and added purified MBP-FLAG-TALE before the PCNA-loading reaction to block PCNA from sliding off.”

Page 8, line 4-6

Inserted following sentence.

“The PCNA-unloading activity of ATAD5-RLC was also observed with longer DNA substrate that contains nine primer-template junctions (Supplementary Fig. 1e).”

Page 8, line 24 ~ Page 9, line 12

Inserted following sentences.

“Because previous studies reported PCNA-unloading activity of RFC¹⁸⁻²¹, we more carefully examined whether RFC could unload PCNA (Supplementary Fig. 1k-n). First, we reduced ATP concentration in PCNA-unloading reaction, since it was reported that high ATP concentration could inhibit PCNA-unloading activity of RFC²⁰. However, the amount of DNA-loaded PCNA was not reduced by RFC at 0.5 mM or 1 mM ATP (Supplementary Fig. 1k). PCNA-unloading activity of ATAD5-RLC was not affected by ATP concentration. Increase of RFC in unloading reaction reduced the amount of DNA-loaded PCNA (Supplementary Fig. 1l). However, PCNA-unloading activity of RFC was significantly lower than that of ATAD5-RLC (compare Supplementary Fig. 1k and Supplementary Fig. 1l). Next, we examined whether the gap-size of DNA affects PCNA-unloading activity of RFC (Supplementary Fig. 1m). We compared PCNA unloading from 10-nucleotide-gap DNA and nicked DNA. RFC unloaded PCNA more efficiently from the nicked DNA. However, even with nicked DNA, PCNA-unloading activity of RFC was significantly lower than that of ATAD5 (Supplementary Fig. 1n). These results confirm that ATAD5-RLC is the most potent PCNA-unloader among clamp-loader complexes.”

Page 20, line 14-16

Inserted following sentence.

“Although RFC could reduce the amount of DNA-loaded PCNA, its PCNA unloading activity is significantly lower than that of ATAD5-RLC (Supplementary Fig. 1k-n).”

Page 21, line 14-18

Inserted following sentence.

“Although RFC could unload PCNA, its PCNA unloading activity is significantly lower than that of ATAD5-RLC (Supplementary Fig. 1k-n). Different from ATAD5-RLC, previous reports showed that RFC-mediated PCNA unloading requires ATP hydrolysis¹⁰. RFC-mediated PCNA unloading was affected by DNA structure, while ATAD5-RLC could unload PCNA regardless of DNA structure (Supplementary Fig. 1h-j, m).”

Page 18, line 4-5

“Our results are the first report to biochemically show that only Elg1-RLC has PCNA-unloading activity among yeast clamp-loader complexes.”

→ “Therefore, Elg1-RLC is the most potent PCNA unloader among yeast clamp loader complexes.”

Major point 4

From the notion that PCNA unloading should be prevented during DNA synthesis DNA to prevent futile PCNA-loading and unloading cycle (Page 8, lines6-7), the authors studied the unloading with DNA substrates harboring a nick or a 11-nucleotide gap, and observed efficient unloading from both DNA. Then, they concluded that DNA structures, on which PCNA remains attached, did not significantly affect PCNA-unloading by ATAD5-RLC (Page 16, lines 7-8). However, these structures will not represent completion of DNA synthesis. Indeed, it has been reported that PCNA unloading by Elg1-RLC required the ligation of Okazaki fragment (Cell rep, 2015, 12, 774). Thus, analysis of PCNA-unloading from DNA, in which the nick has been ligated after PCNA loading, is necessary to test the structural

differences of replicating and replicated DNA.

We thank you for the suggestion. To address this question, we added T4 DNA ligase after PCNA loading to nicked DNA (Supplementary Fig. 1j). The nick contains 5'-phosphate for ligation. Treatment of T4 DNA ligase did not affect PCNA unloading efficiency of ATAD5-RLC. We believe that this result further strengthens our argument that DNA structure does not affect PCNA-unloading activity of ATAD5-RLC.

According to this result, we revised our manuscript as below.

Page 8, line 12-23

“Therefore, we tested PCNA unloading from two DNA substrates mimicking steps of lagging strand synthesis (Supplementary Fig. 1f and g). To maximize the effect of the structural difference, we prepared short 80-mer DNA substrates. Each substrate was prepared by annealing three oligonucleotides (Supplementary Fig. 1f). The 5'-biotinylated 18-mer was annealed to 80-mer DNA. In each substrate, another oligo was annealed to the 80-mer DNA at downstream of the 18-mer to generate a 1- or 11-nucleotide gap, respectively. PCNA should be loaded to the 3' end of the 18-mer in the middle of the substrate. Because the substrates were short, we added a TALE-binding sequence to the opposite side of biotinylation to block PCNA from sliding off, and added purified MBP-FLAG-TALE before the PCNA-loading reaction. Contrary to the assumption, ATAD5-RLC unloaded PCNA from different substrates with similar efficiency (Supplementary Fig. 1g). This result implies that the size of single-stranded gap in DNA replication intermediates does not affect PCNA unloading activity of ATAD5-RLC and there might be other mechanisms to regulate PCNA unloading during DNA synthesis.”

→ “Therefore, we tested PCNA unloading from DNA substrates mimicking steps of lagging strand synthesis (Supplementary Fig. 1h-j). We first examined whether the gap-size of DNA affected PCNA unloading. We prepared two short 80-mer DNA substrates, which contained

1 or 11 nucleotides gap, respectively (Supplementary Fig. 1h). Slide-off of PCNA was blocked by TALE binding. ATAD5-RLC unloaded PCNA from those two substrates with similar efficiency (Supplementary Fig. 1i). To examine whether fully duplexed DNA enhances PCNA unloading by ATAD5-RLC, we prepared DNA substrate containing single nick (Supplement Fig. 1j). After PCNA loading, T4 DNA ligase was added to the reaction to ligate DNA fragments. Addition of T4 DNA ligase did not affect PCNA-unloading activity of ATAD5-RLC. These results imply that the DNA structure on which PCNA stays does not affect PCNA-unloading activity of ATAD5-RLC and there might be other mechanisms to regulate PCNA unloading during DNA synthesis.”

Major point 5

The authors found that Fen1 could inhibit the PCNA unloading and suggested that Fen1 inhibits PCNA unloading during DNA replication (Page 18, lines 2-3). The similar mechanism by DNA polymerase δ for PCNA unloading by RFC has been published by Hedglin et al. (eLife 2: e00278). Thus, competition of PCNA binding by several PCNA interacting proteins and loader complexes will be a general mechanism to regulate PCNA cycling, not only for ATAD5-RLC. This point should be considered to revise this discussion.

As reviewer pointed out, it was reported that RFC-mediated PCNA unloading could be inhibited by polymerase δ . We agree that competition between clamp-loader complexes and PCNA-interacting proteins could serve as a mechanism to regulate PCNA cycling. We also examined the effect of polymerase δ on PCNA unloading in the revised manuscript (Fig. 6f). We found that budding yeast polymerase δ could inhibit PCNA unloading by Elg1-RLC. We cited the previous report and revised our discussion as reviewer suggested.

Page 19, line 4-6

“Among these, Fen1, which cleaves flaps during the maturation of Okazaki fragments, strongly inhibited the unloading reaction. Ligase I also partially inhibited PCNA unloading.”

→ “Among these, Fen1, which cleaves flaps during the maturation of Okazaki fragments, strongly inhibited the unloading reaction. RPA, Ligase I and polymerase δ also partially inhibited PCNA unloading.”

Page 22, line 20-23

“We found that yeast Fen1 could inhibit PCNA unloading by Elg1-RLC through its binding to PCNA *in vitro* (Fig. 6). Although inhibition was less efficient than Fen1, Ligase 1 also inhibited the PCNA unloading reaction *in vitro*.”

→ “We found that yeast Fen1 could inhibit PCNA unloading by Elg1-RLC through its binding to PCNA *in vitro* (Fig. 6). Although less effective than Fen1, other replication enzymes – Ligase 1, polymerase δ , and RPA – also inhibited the PCNA unloading reaction *in vitro*.”

Page 23, line 10-12

Inserted following sentences:

“It was reported that RFC-mediated PCNA unloading is also inhibited by polymerase δ ²⁰. Therefore, competition between clamp-loader complexes and PCNA-interacting proteins might be a general mechanism for PCNA cycling.”

Minor point 1

In Fig. 1b, Fig. 2e, f, Fig. 3b, the leftmost lanes are the controls without ATAD5 depletion.

Surprisingly, no chromatin-bound PCNA was detected in these untreated 293T cells, though a significant population of the cells should be in S phase, in which PCNA should exist on the replicating chromatin. Indeed, chromatin-bound PCNA was detected in the controls in Fig. S2f and Fig. S5d. These inconsistencies should be explained from a technical view.

Because PCNA was highly accumulated on the ATAD5-depleted chromatin, PCNA signal on the control (undepleted) chromatin was often weakened to the level that was hard to be detected. That does not mean that PCNA is absent on the control chromatin. We repeated experiments and presented blots that are suitable for the analysis of PCNA-level on the chromatin.

Minor point 2

PCNA unloading by yeast Elg1-RLC seems to be less efficient than by ATAD4-RLC, as the used concentrations were 75-150nM for the former and 0.15-0.6nM for the latter. This point should be explained clearly. I am afraid that this might occur by used of truncated version of ATAD5-RLC, in which some negative regulatory domains would be removed from the intact one.

As we described in the response to Major point 2 of reviewer #2, full-length ATAD5-RLC and N-terminal deleted ATAD5-RLC unloaded PCNA with similar efficiency. Furthermore, we prepared N-terminal deleted Elg1-RLC and compared its unloading activity with full-length Elg1-RLC (Fig. 6d and e). N-terminal deletion of Elg1 did not enhance its PCNA unloading activity. Therefore, the N-terminal regions of both ATAD5 and Elg1 are not essential for their unloading activity. Although Elg1-RLC is less efficient compared to ATAD5-RLC, this difference is not caused by the presence of N-terminal domain. We assume enzymes originated from different species might have different enzymatic efficiencies.

We revised our manuscript as below.

Page 10, line 23 ~ Page 11, line 3

Inserted following sentences.

“To more clearly define functional contributions of each ATAD5 domain for PCNA unloading, we purified ATAD5 (Δ N692) alone and full-length ATAD5-RLC using the same system

(Supplementary Fig. 2i) and examine their PCNA-unloading activity (Supplementary Fig.2j). ATAD5 (Δ N692) alone did not show PCNA-unloading activity. PCNA unloading activity of full-length ATAD5-RLC and ATAD5 (Δ N692)-RLC were similar. Therefore, N-terminal domain of ATAD5 is not essential for PCNA unloading and PCNA unloading requires ATAD5-RLC, not ATAD5 alone.”

Page 18, line 12-16

Inserted following sentences.

“We then prepared N-terminal or C-terminal deletion mutants of Elg1 (Fig. 6d). Deletion of N-terminal 215 amino acids of Elg1 did not affect PCNA-unloading activity of Elg1-RLC (Fig. 6e). Therefore, although Elg1-RLC is less efficient compared to human ATAD5-RLC, it is not due to the presence of N-terminal domain (Fig. 1f and Fig. 6a).”

Minor point 3

Images of H3 in Fig. 1b were too dark to evaluate the amounts of the chromatin fraction. Use an appropriate exposure for the data.

We repeated experiments and presented data that are suitable for the evaluation. Lamin B1 was used as a control for chromatin fraction. We think that we could evaluate the amount of chromatin proteins more accurately from these new results.

Minor point 4

Numbers of primers annealed with 5'-biotinylated single stranded DNA substrate were described as “ten” at Page 7 line16 and “nine” at page 29 line 16. Indicate a precise number there. It is also necessary to describe their sequences in the supplementary information.

We apologize for the error. Nine is correct number. As reviewer suggested, we included sequence information of the DNA in supplementary information.

Minor point 5

At Page 9, lines 19-20, "ATAD5 (693-1719) bound to a lesser amount of RFC2–5 compared to ATAD5 (693-1844)" and Page 17, lines 9-10 "Elg1 (Δ C756)-RLC showed a sub-stoichiometric amount of Rfc2–5". I could not agree with these conclusions only from the images of Fig. S2c and Fig. S6f. Indicate more apparent data.

In the case of ATAD5, C-terminal deletion reduced the stability of ATAD5 in insect cell system. Therefore, we took different approaches to purify ATAD5 (Δ N692)-RLC and ATAD5 (693-1719). We purified those proteins from the combination of yeast and bacterial expression system as described in the response to major point 2 of reviewer #2 (Supplementary Fig. 2g). This new result more clearly shows that ATAD5 (693-1719) is defective for RFC2-5 binding.

During revision process, we found that the deletion of C-terminal 36 amino acids, Elg1 (Δ C756), was not enough to reduce its binding to Rfc2-5. We prepared two more C-terminal deletion mutants of Elg1, Elg1 (Δ C684) and Elg1 (Δ C536), and examined their Rfc2-5 binding (Fig. 6d). Our new data clearly shows that C-terminal 256 amino acids deletion made Elg1 defective in Rfc2-5 binding and PCNA unloading (Fig. 6d and e). Therefore, Elg1 (536-683) is crucial for Rfc2-5 binding. We apologize for presenting incorrect information in our initial manuscript. We revised our manuscript according to our new results as below.

Page 10, line 18-23

"Next, we purified an RLC-formation-defective mutant (693-1719) of ATAD5 (Supplementary Fig. 2c) and examined its PCNA-unloading activity *in vitro* (Fig. 2g). ATAD5 (693-1719) bound to a lesser amount of RFC2–5 compared to ATAD5 (693-1844). ATAD5 (693-1719) could not unload PCNA from the DNA substrate."

→ “Next, we purified an RLC-formation-defective mutant (693-1719) of ATAD5 (Supplementary Fig. 2f). Because ATAD5 (693-1719) was less stable in insect cells, we also purified ATAD5 (693-1719) and ATAD5 (Δ N692)-RLC using the combination of yeast and bacterial expression system (Supplementary Fig. 2g). ATAD5 (693-1719) bound to a lesser amount of RFC2–5 compared to ATAD5 (693-1844) and was defective in PCNA unloading in vitro (Fig. 2g and Supplementary Fig. 2h).”

Page 18, line 16-18

“We then prepared a C-terminal deletion mutant of Elg1, in which C-terminal RBM was removed (Δ C756). Elg1 (Δ C756)-RLC showed a sub-stoichiometric amount of Rfc2–5 and was defective in PCNA unloading. Furthermore, ATP- γ -S interfered with efficient PCNA unloading (Fig. 6d).”

→ “Elg1 (Δ C536)-RLC did not bind to Rfc2–5 and was defective in PCNA unloading compared to wild type or Elg1 (Δ C684) (Fig. 6d-e). These results suggest that 215th ~ 683rd amino-acid region of Elg1 is crucial for PCNA unloading activity.”

Minor point 6

At Page 16, lines 10-11, “To examine this possibility, we took advantage of the in vitro DNA replication system of budding yeast” will be “To examine this possibility, we took advantage of budding yeast replication proteins”

We revised the sentence as reviewer suggested.

Minor point 7

At Page 18, lines 1-2, “By contrast, a mutation in the nuclease motif of Fen1 (CAT) did not inhibit PCNA unloading” should be “By contrast, a mutation in the nuclease motif of Fen1 (CAT) did inhibit PCNA unloading”.

We corrected this sentence as reviewer suggested.

Minor point 8

“N693” in Fig. S5d has not been described in this text. I guess this will be the ATAD5 fragment from 1 to 693 aa with some protein tag, but it should be specified.

We included the information about ATAD5 (N693) in the legend of Supplementary Fig. 5f.

“ATAD5 (N693) and ATAD5 (Δ N692) were transiently expressed in ATAD5-depleted cells.”

→ “N-terminal FLAG-tagged 1st ~ 693rd amino-acid region of ATAD5 (ATAD5 (N693)) or 693rd ~ 1844th amino-acid region of ATAD5 (ATAD5 (Δ N692)) was transiently expressed in ATAD5-depleted cells.”

Reviewer #3.

Review of manuscript: “Regulation of PCNA cycling on replicating DNA by RFC and RFC-like complexes” by Mi-Sun Kang et al.

In this manuscript the authors take a biochemical approach to investigate the relative roles of RFC and RFC-like complexes (RLCs) in PCNA metabolism. They show that whereas RFC and the CTF18 RLC load PCNA, they do not unload it from chromatin. The Rad24 RLC loads the 9-1-1 complex, and the ATAD5-Elg1 RLC is the only one that unloads PCNA from the DNA. By carrying out single-molecule measurements, they arrive at the conclusion that loading by RFC can be divided into three steps, whereas unloading by ATAD5 shows only two. Unloading was independent of the DNA structure, and was not affected by PCNA modifications.

In general, the analysis of these complexes is interesting and important and the data presented are sound. The paper will be of interest to a wide readership. I have only a few queries, and many minor corrections, mainly of the English.

Major:

1) None of the figures includes quantitation. This is unacceptable, particularly for experiments such as PCNA loading/unloading in which the authors derive conclusions based on quantities. Some of the “control proteins” that could be used as reference for quantitation are extremely overexposed, and cannot serve as a measure for equal loading.

2) The role played by ATP in the activity of the ATD5-Elg1-RLC is confusing to this referee. The authors conclude, on one hand, that the WalkerA and WalkerB motifs are essential for PCNA unloading activity (the mutations accumulate PCNA at the

same level as cells depleted of ATAD5). Moreover, defects in the ATPase cause reduced EdU incorporation in vivo in dividing cells (Figure 3). On the other hand, the authors show that ATP-Gamma-S only reduces activity and can be easily overcome by additional ATAD5 complex (Figure 3e) and FRET experiments show that ATP-Gamma-S did not affect the high-FRET state. The authors then say that ATP hydrolysis may not be required to bring down PCNA, only to dissociate ATAD5 from it. But how do they then explain the results obtained by the Walker A/B mutants? If ATP hydrolysis was only a post-unloading step, the mutants should still be able to unload PCNA from chromatin.

3) The effects of CM sequences on the interaction with the small Rfc subunits and on PCNA unloading are not internally consistent, and remain unexplained.

4) To study the possible role of ATAD5 independently of the small Rfc subunits, instead of mutating the interphase, the authors can attempt unloading assays in the absence of the small subunits.

Minor:

4) What is the nature of the relatively strong bands of about 130 MDa in Figure 1c, or the asterisk-marked band in Suppl. Figure 2C?

5) The model represented by Suppl. Figure 3F cannot be understood from the drawing nor from the legend.

6) There is a need for English editing throughout the paper. Here are some examples:

Page 4, line 17: “experiments”

Page 4, line 18: “experiments”

Page 4, last line: “mammalian cells are sensitive”

Legend of Figure 1: “is conferred by the C-terminal domain”

Page 7, line 12: “we purified the N-terminal-deleted...”

Page 9, line 16: delete “were”

Page 10, line 12: form

Page 10, 4 lines before end: quote Supplementary Figure 3.

Page 17, line 7: something is wrong with the sentence: “those Walker A and Walker

B...”

Major point 1

None of the figures includes quantitation. This is unacceptable, particularly for experiments such as PCNA loading/unloading in which the authors derive conclusions based on quantities. Some of the “control proteins” that could be used as reference for quantitation are extremely overexposed, and cannot serve as a measure for equal loading.

We apologize for the lack of quantitative data in our initial manuscript. We included quantitative results for the analysis of chromatin bound PCNA and *in vitro* PCNA-unloading

experiments in our revised manuscript. To overcome the issue of overexposed “control proteins”, we repeated experiments and present new data showing better control proteins – Lamin B1 for Fig. 1b, Fig. 2e, f, Fig. 3b, Supplementary Fig. 2e, Supplementary Fig. 3c and Supplementary Fig. 5f. We believe that our new data with quantitative results significantly strengthen our arguments.

Major point 2

The role played by ATP in the activity of the ATD5-Elg1-RLC is confusing to this referee. The authors conclude, on one hand, that the Walker A and Walker B motifs are essential for PCNA unloading activity (the mutations accumulate PCNA at the same level as cells depleted of ATD5). Moreover, defects in the ATPase cause reduced EdU incorporation in vivo in dividing cells (Figure 3). On the other hand, the authors show that ATP-Gamma-S only reduces activity and can be easily overcome by additional ATAD5 complex (Figure 3e) and FRET experiments show that ATP-Gamma-S did not affect the high-FRET state. The authors then say that ATP hydrolysis may not be required to bring down PCNA, only to dissociate ATAD5 from it. But how do they then explain the results obtained by the Walker A/B mutants? If ATP hydrolysis was only a post-unloading step, the mutants should still be able to unload PCNA from chromatin.

As reviewer pointed out, we presented two data sets. One is that ATPase-motif mutants (Fig. 3 and Supplementary Fig. 3) are defective in PCNA unloading both *in vitro* and in cells. The other is that PCNA unloading by ATAD5-RLC occurs before ATP hydrolysis (Fig. 3 and Fig. 4). As we discussed in the manuscript, we think that ATP hydrolysis might be required for the separation of ATAD5-RLC from PCNA after its release from DNA for the next PCNA-unloading cycle. Therefore, ATPase-motif mutants might fail to reduce the level of chromatin bound PCNA because mutant ATAD5-RLC cannot be recycled due to ATP binding/hydrolysis defect. Another possibility is that ATPase-motif mutants might be in a

conformation that is defective in PCNA unloading. Because ATPase-motif are located between ATAD5 and RFC2-5, we think that mutation in ATPase-motif could cause structural distortion in addition to ATP binding/hydrolysis defect. We describe these possibilities more carefully in text as below.

Page 21, line 11-14

Inserted following sentences.

“ATPase-motif mutants might fail to reduce the level of chromatin bound PCNA because mutant ATAD5-RLC cannot be recycled due to ATP binding/hydrolysis defect. Otherwise, ATPase-motif mutants might take a conformation that is defective in PCNA unloading.”

Major point 3

The effects of CM sequences on the interaction with the small Rfc subunits and on PCNA unloading are not internally consistent, and remain unexplained.

We prepared four CM mutants and tested their RFC2-5 binding and PCNA-unloading activities. CM1 mutant failed to bind RFC2-5 and were also defective in PCNA unloading (Fig 2d and f). As reviewer pointed out, other CM mutants (CM2~4) could interact with RFC2-5 but showed different levels of PCNA unloading defect. CM3 and CM4 are functional for PCNA unloading, but CM2 is partially defective in PCNA unloading. We think that CM2 mutations might cause inappropriate binding of RFC2-5 that compromise enzymatic activity. We describe this possibility in detail in the revised manuscript.

Page 10, line 8-17

“Interestingly, the CM2 mutation of ATAD5 (S¹⁷²⁵KALE¹⁷²⁹) resulted in partial PCNA unloading defect, although the CM2 mutant binds well to RFC2-5 (Fig. 2f). Furthermore, when the RBM of ATAD5 (residues 1601–1844) was swapped with that of RFC1 (residues

835–1147), PCNA-unloading activity was significantly reduced (Supplementary Fig. 2d-f), although the swap mutant still formed a complex with RFC2–5. This result suggests that the RBM of ATAD5 was not only required for ATAD5-RLC complex formation, but for forming the active conformation of ATAD5-RLC to unload PCNA.”

→ “Although other CM mutants (CM2-CM4) were not defective in RFC2-5 binding (Fig. 2d), those mutants showed different levels of PCNA-unloading defects (Fig. 2f). These results suggest that those mutations in RBM might induce conformational changes of ATAD5-RLC, which compromise its catalytic activity. Furthermore, when the RBM of ATAD5 (residues 1601–1844) was swapped with that of RFC1 (residues 835–1147), PCNA-unloading activity was significantly reduced (Supplementary Fig. 2c-e), although the swap mutant still formed a complex with RFC2–5. This result suggests that the RBM of ATAD5 was not only required for ATAD5-RLC complex formation, but for forming the active conformation of ATAD5-RLC to unload PCNA.”

Major point 4

To study the possible role of ATAD5 independently of the small Rfc subunits, instead of mutating the interphase, the authors can attempt unloading assays in the absence of the small subunits.

As reviewer suggested, we purified ATAD5 (Δ N692) without RFC2-5 (Supplementary Fig. 2i). Since ATAD5 (Δ N692) was not stably expressed in insect cells, we adopted yeast expression system. We showed that ATAD5 (Δ N692) alone did not unload PCNA from DNA (Supplementary Fig. 2j). We believe that this result strengthens our argument that ATAD5 should form a complex with RFC2-5 to unload PCNA.

We revised our manuscript as below.

Page 10, line 23 ~ Page 11, line 3

“Therefore, PCNA unloading requires the ATAD5-RLC complex, not just ATAD5 alone.”

→ “To more clearly define functional contributions of each ATAD5 domain for PCNA unloading, we purified ATAD5 (Δ N692) alone and full-length ATAD5-RLC using the same system (Supplementary Fig. 2i) and examine their PCNA-unloading activity (Supplementary Fig.2j). ATAD5 (Δ N692) alone did not show PCNA-unloading activity. PCNA unloading activity of full-length ATAD5-RLC and ATAD5 (Δ N692)-RLC were similar. Therefore, N-terminal domain of ATAD5 is not essential for PCNA unloading and PCNA unloading requires ATAD5-RLC, not ATAD5 alone.”

Minor point 1

What is the nature of the relatively strong bands of about 130 MDa in Figure 1c, or the asterisk-marked band in Suppl. Figure 2C?

We think that those bands represent partially degraded forms of ATAD5 (Δ N692). ATAD5-RLC purified from yeast expression system also contains small amount of proteins around similar molecular weights. We think that the presence of those bands do not significantly affect our analysis. In the case of Supplementary Fig. 2c (now Supplementary Fig. 2f), asterisk-marked band seems to be originated from insect cell lysate. We prepared better ATAD5 (Δ N692) as shown in Supplementary Fig. 2g, to analyze its PCNA-unloading activity.

Minor point 2

The model represented by Suppl. Figure 3F cannot be understood from the drawing nor from the legend.

We apologize for the confusion. We removed the cartoon from our manuscript.

Minor point 3

There is a need for English editing throughout the paper. Here are some examples:

Page 4, line 17: "experiments"

Page 4, line 18: "experiments"

Page 4, last line: "mammalian cells are sensitive"

Legend of Figure 1: "is conferred by the C-terminal domain"

Page 7, line 12: "we purified the N-terminal-deleted..."

Page 9, line 16: delete "were"

Page 10, line 12: form

Page 10, 4 lines before end: quote Supplementary Figure 3.

Page 17, line 7: something is wrong with the sentence: "those Walker A and Walker B..."

We apologize for typographical and grammatical errors. We fixed those errors in our revised manuscript.

Reviewer #4

The manuscript by Kang et al. aims to dissect how distinct clamp loaders/unloaders mediate the lifetime of PCNA on DNA. The manuscript encompasses a very large body of biochemical work on the ATAD5-RLC. Detailed studies addressing the critical questions in this field have remain elusive, largely due to the difficulty in obtaining ATAD5-RLC for biochemical experiments. The authors have made a significant leap in purifying a truncated form of human ATAD5-RLC for biochemical experiments and some of the work presented characterizes how the ATAD5 subunit is incorporated into a functional RLC complex with unloading activity. However, at this stage, the major findings of the manuscript are limited in their novelty and scope (see details below) and, hence, do not warrant publication in Nature Communications in the current format.

Major Concerns:

1. Based largely on the results presented in Figure 1E, the authors argue that ATAD5-RLC is the only PCNA unloader. This contradicts a large body of work from many labs (O'Donnell, Hurwitz, Benkovic, Fanning, etc) which independently demonstrated that, in addition to loading PCNA onto DNA, human RFC also unloads PCNA from DNA and such activity is robust. This work was neither cited nor discussed in the manuscript. The discrepancy between the current manuscript and the previous work may lie in the experimental design.

The way the substrate is depicted in Figure 1D, there is nothing to prevent PCNA from sliding off the 3' end of the 5'-biotinylated DNA substrate due to the absence of physical blocks (RPA, secondary DNA structures such as hairpins, etc.). Work from the van Oijen and Walter labs demonstrated that human PCNA rapidly diffuses along kilobases of DNA in a random manner (JBC 2009). After a 30 minute incubation at 37°C on a 1.4 kb DNA substrate, I would envision that a significant portion of the PCNA-bound DNA would lose its PCNA during the incubation simply by random diffusion (i.e., in a manner independent of any unloading activity). This behavior was not observed, suggesting that the DNA substrate may contain prominent intramolecular secondary structures. RFC-catalyzed unloading of PCNA is substrate-dependent. Furthermore, previous work from the Benkovic lab revealed that human RFC-catalyzed unloading of human PCNA from DNA is inhibited at ATP concentrations greater than 1 mM.

For instance, increasing ATP concentration from 1 mM to 5 mM ATP inhibits RFC-catalyzed unloading ~4-fold (eLife 2013). The unloading experiments depicted in Figure 1E were carried out at 6 mM ATP (well above physiological levels) where RFC-catalyzed unloading of PCNA is significantly inhibited. Thus, RFC unloading activity is inhibited under the experimental conditions and possibly by the DNA substrate itself. As is, any conclusion/suggestion comparing the relative PCNA unloading activities of RFC and ATAD5 are restricted to the experiment described in Figure 1D – E for the 1.4 kb DNA substrate and are not generally applicable.

2. The smFRET results described in Figure 4 pertaining to RFC-catalyzed loading of PCNA confirm previous findings from ensemble FRET measurements on human RFC by the Benkovic lab (eLife, 2013, Biochemistry 2017). Namely, human PCNA loading occurs through two intermediate states on DNA, separated by ATP hydrolysis and, hence, human RFC-catalyzed loading of human PCNA stalls at the first intermediate state in the presence of ATP γ S.

3. The results described in Figure 5 and 6 pertaining to unloading of PCNA by ATAD5/ELG1-RLC were previously observed for unloading of human PCNA by human RFC; work from the Benkovic lab demonstrated that the robust PCNA unloading activity of human RFC is independent of PCNA monoubiquitination (PNAS 2016) and inhibited by DNA replication proteins that interact with PCNA through PIP boxes (eLife 2013). Hence, the only difference between unloading of PCNA by RFC and ATAD5-RLC is that RFC-catalyzed unloading of PCNA is dependent on ATP hydrolysis whereas ATP hydrolysis is not required for ATAD5-RLC catalyzed unloading of PCNA, as indicated in the current manuscript.

The comparative unloading experiments should be repeated at ATP concentrations < 1 mM on a minimal DNA substrate (similar to the one described in Supplementary Figure 1F) where both RFC and ATAD5-RLC catalyzed unloading of PCNA are observed. The unloading activities should be quantitatively assessed. This DNA substrate should then be systematically altered (i.e., increasing/decreasing the primer lengths, etc.) and the relative unloading activities of RFC and ATAD5-RLC quantitatively compared to decipher where the differences in unloading activities, if any, lie.

Minor Concerns

1. The authors claim that PCNA unloading requires the ATAD5-RLC complex, not just ATAD5 alone. This conclusion cannot be reached without an experiment where PCNA unloading is monitored in the presence of only the wild-type ATAD5 subunit.

2. Functionality of the Cy3-labeled PCNA. The ability of PCNA to interact with other proteins, be loaded onto DNA, etc. is sensitive to amino acid mutations and modifications. To my knowledge, this is the first report where the native, exposed cysteines of PCNA were labeled with Cy3. It is customary to demonstrate that such labeling does not affect the ability of PCNA to function properly. Also, without correcting for the absorbance of Cy3 at 280 nm or determining the concentration of the protein via Bradford, the percent labeling is likely an underestimate.

Major point 1

Based largely on the results presented in Figure 1E, the authors argue that ATAD5-RLC is the only PCNA unloader. This contradicts a large body of work from many labs (O'Donnell, Hurwitz, Benkovic, Fanning, etc) which independently demonstrated that, in addition to loading PCNA onto DNA, human RFC also unloads PCNA from DNA and such activity is robust. This work was neither cited nor discussed in the manuscript. The discrepancy between the current manuscript and the previous work may lie in the experimental design. The way the substrate is depicted in Figure 1D, there is nothing to prevent PCNA from sliding off the 3'end of the 5'-biotinylated DNA substrate due to the absence of physical blocks (RPA, secondary DNA structures such as hairpins, etc.). Work from the van Oijen and Walter labs demonstrated that human PCNA rapidly diffuses along kilobases of DNA in a random manner (JBC 2009). After a 30 minute incubation at 37oC on a 1.4 kb DNA substrate, I would envision that a significant portion of the PCNA-bound DNA would lose its PCNA during the incubation simply by random diffusion (i.e., in a manner independent of any unloading activity). This behavior was not observed, suggesting that the DNA substrate may contain prominent intramolecular secondary structures. RFC-catalyzed unloading of PCNA is substrate-dependent. Furthermore, previous work from the Benkovic lab revealed that human RFC-catalyzed unloading of human PCNA from DNA is inhibited at ATP concentrations greater than 1 mM.

For instance, increasing ATP concentration from 1 mM to 5 mM ATP inhibits RFC-catalyzed unloading ~4-fold (eLife 2013). The unloading experiments depicted in Figure 1E were carried out at 6 mM ATP (well above physiological levels) where RFC-catalyzed unloading of PCNA is significantly inhibited. Thus, RFC unloading activity is inhibited under the experimental conditions and possibly by the DNA substrate itself. As is, any conclusion/suggestion comparing the relative PCNA unloading activities of RFC and ATAD5

are restricted to the experiment described in Figure 1D – E for the 1.4 kb DNA substrate and are not generally applicable.

We would like to answer this comment below, combined with Major point 3.

Major point 2

The smFRET results described in Figure 4 pertaining to RFC-catalyzed loading of PCNA confirm previous findings from ensemble FRET measurements on human RFC by the Benkovic lab (eLife, 2013, Biochemistry 2017). Namely, human PCNA loading occurs through two intermediate states on DNA, separated by ATP hydrolysis and, hence, human RFC-catalyzed loading of human PCNA stalls at the first intermediate state in the presence of ATP γ S.

We acknowledge and appreciate that the stopped-flow ensemble FRET measurements showed the existence of multiple conformational steps separated by ATP hydrolysis in the course of PCNA loading (Hedglin *et al.*, eLife 2013). Single molecule FRET (smFRET) measurements clarify several critical features of the process, by its advantage of tracing the conformational changes of each molecule. As stated by Hedglin and colleagues, observation of multi-exponential time trace in stopped-flow measurements, in general, does not always clarify if it is due to either multistep process or the presence of more than one population (or pathway) of the complex. Our smFRET measurements showed that the conformational changes occurred in succession in each complex, unambiguously revealing that each PCNA clamp goes through multi-step conformational transitions.

Secondly, from the stopped-flow measurements with ATP- γ -S, it was not clear if ATP- γ -S leads to the same FRET state as that of the first intermediate with ATP, possibly because multiple processes were occurring simultaneously (RFC-PCNA loading, diffusion, multiple binding, dissociation, etc.). smFRET measurements clearly showed that loading with ATP- γ -

S led to a single FRET state whose level matched that of the first intermediate in loading with ATP (LI1).

In addition, our smFRET measurements showed that ATAD5-RLC-driven unloading process occurred through a conformational intermediate (UI) that has the same FRET level as the second intermediate observed during loading (LI2), which would not be trivial to observe from ensemble measurements when multiple processes with different lifetimes are occurring simultaneously (e.g. unsuccessful unloading trials as observed in our data). Analyses of smFRET data also revealed the dwell time distributions, suggesting that LI2 and UI are intrinsically distinct states though they exhibit the same FRET level.

In recognition of the good agreement with the results from preceding stopped-flow measurements, we revised the text as below citing the relevant work.

Page 15, line 17-19

“This result indicated that LI2 was an intermediate that formed after ATP hydrolysis.”

→ “This result indicates that LI1 and LI2 are two intermediate conformational steps separated by ATP hydrolysis, consistent with what was suggested from stopped-flow FRET measurements (Hedglin et al., eLife 2013).”

Major point 3

The results described in Figure 5 and 6 pertaining to unloading of PCNA by ATAD5/ELG1-RLC were previously observed for unloading of human PCNA by human RFC; work from the Benkovic lab demonstrated that the robust PCNA unloading activity of human RFC is independent of PCNA monoubiquitination (PNAS 2016) and inhibited by DNA replication proteins that interact with PCNA through PIP boxes (eLife 2013). Hence, the only difference between unloading of PCNA by RFC and ATD5-RLC is that RFC-catalyzed unloading of PCNA is dependent on ATP hydrolysis whereas ATP hydrolysis is not required for ATAD5-RLC catalyzed unloading of PCNA, as indicated in the current manuscript.

The comparative unloading experiments should be repeated at ATP concentrations < 1 mM on a minimal DNA substrate (similar to the one described in Supplementary Figure 1F) where both RFC and ATAD5-RLC catalyzed unloading of PCNA are observed. The unloading activities should be quantitatively assessed. This DNA substrate should then be systematically altered (i.e., increasing/decreasing the primer lengths, etc.) and the relative unloading activities of RFC and ATAD5-RLC quantitatively compared to decipher where the differences in unloading activities, if any, lie.

We appreciate for the detailed suggestions by this reviewer. As reviewer pointed out, there are previous reports showing PCNA-unloading activity of RFC. We apologize that we did not cite those reports in our initial manuscript. To understand discrepancy between previous reports and our results about RFC-mediated PCNA unloading, we designed and performed experiments as the reviewer suggested. First, we reduced ATP concentration in PCNA-unloading reaction to 0.5 - 1 mM to examine the possibility that high ATP concentration might inhibit PCNA unloading by RFC (Supplementary Fig. 1k). With 1.4 Kbps DNA, we observed reduction of PCNA by simple diffusion. However, PCNA remained on DNA enough to monitor its unloading by ATAD5-RLC. In this experiment, we used 130-mer DNA that contains 10-nucleotides gap instead of 1.4 Kbps DNA, as the reviewer suggested. Bead-unattached side of the DNA was blocked by TALE-binding protein to prevent slide-off of PCNA. TALE binding significantly increased the amount of DNA-loaded PCNA. Same as our initial results, RFC did not reduce the amount of DNA-loaded PCNA. ATAD5-RLC efficiently unloaded PCNA independent of ATP concentration. We repeated experiments in Fig. 1e with 130-mer gapped DNA and 1 mM ATP concentration. This result also confirms our previous observation that only ATAD5-RLC showed PCNA unloading activity. RFC did not show PCNA unloading activity in single molecule experiment, either (Supplementary Fig. 4f). However, when we increased the concentration of RFC in PCNA-unloading reaction, we could observe the reduction of DNA-loaded PCNA (Supplementary Fig. 1l). But, PCNA

unloading activity of ATAD5-RLC is significantly higher than that of RFC (compare Supplementary Fig. 1k and Supplementary Fig. 1l). Next, we examined whether the gap-size of DNA affect RFC-mediated PCNA unloading (Supplementary Fig. 1l). We compared 10-nucleotide-gap DNA and nicked DNA for PCNA unloading by RFC. RFC reduced DNA-loaded PCNA efficiently from nicked DNA compared to 10-nucleotide gapped DNA. Even with nicked DNA, PCNA unloading activity of RFC was significantly lower than that of ATAD5 (Supplementary Fig. 1n). These results confirms that ATAD5-RLC is the most potent PCNA-unloader among clamp-loader complexes. Furthermore, our results reveal distinct features of RFC compared to ATAD5-RLC in PCNA-unloading reaction. First, although RFC possesses PCNA unloading activity as previously reported, its PCNA unloading activity is significantly lower than that of ATAD5-RLC. Second, RFC more efficiently unloads PCNA from nicked DNA compared to 10-nucleotide gapped DNA. ATAD5-RLC unloads PCNA from various DNA substrates with similar efficiency (Supplementary Fig. 1h-j). RFC might function as a secondary PCNA-unloader during DNA replication. We revised our manuscript according to these new findings as below.

Page 2, line 7-8

“ATAD5-RLC uniquely possesses PCNA unloading activity while RFC, CTF18-RLC, and RAD17-RLC catalyze clamp loading only.”

→ “ATAD5-RLC possesses the most potent PCNA unloading activity while RFC, CTF18-RLC, and RAD17-RLC mainly catalyze clamp loading.”

Page 6, line 9

“ATAD5-RLC is a specific PCNA unloader”

→ “ATAD5-RLC is a potent PCNA unloader”

Page 7, line 14-18

“We prepared 5'-biotinylated single-stranded DNA substrate (1.4 Kbs) on which ten 20-mer oligonucleotides were annealed with ~150-nucleotide gaps between them. This substrate

DNA mimics primer-template junctions. The substrate DNA was then attached to the streptavidin-coated magnetic beads.”

→ “We prepared 130-mer DNA that contains 10 nucleotides gap. One end of the 130-mer DNA was biotinylated and attached to the streptavidin-coated magnetic beads. Because the substrate-DNA is short, we added a TALE-binding sequence to the opposite side of biotinylation and added purified MBP-FLAG-TALE before the PCNA-loading reaction to block PCNA from sliding off.”

Page 8, line 4-6

Inserted following sentence.

“The PCNA-unloading activity of ATAD5-RLC was also observed with longer DNA substrate that contains nine primer-template junctions (Supplementary Fig. 1e).”

Page 8, line 24 ~ Page 9, line 12

Inserted following sentences.

“Because previous studies reported PCNA-unloading activity of RFC¹⁸⁻²¹, we more carefully examined whether RFC could unload PCNA (Supplementary Fig. 1k-n). First, we reduced the ATP concentration in PCNA-unloading reaction, since it was reported that high ATP concentration could inhibit PCNA-unloading activity of RFC²⁰. However, the amount of DNA-loaded PCNA was not reduced by RFC at 0.5 mM or 1 mM ATP (Supplementary Fig. 1k). PCNA-unloading activity of ATAD5-RLC was not affected by ATP concentration. Increase of RFC in unloading reaction reduced the amount of DNA-loaded PCNA (Supplementary Fig. 1l). However, PCNA unloading activity of RFC was significantly lower than that of ATAD5-RLC (compare Supplementary Fig. 1k and Supplementary Fig. 1l). Next, we examined whether gap-size of DNA affects PCNA-unloading activity of RFC (Supplementary Fig. 1m). We compared PCNA unloading from 10-nucleotide-gap DNA and nicked DNA. RFC unloaded PCNA more efficiently from the nicked DNA. However, even with nicked DNA, PCNA unloading activity of RFC is significantly lower than that of ATAD5 (Supplementary

Fig. 1n). These results confirm that ATAD5-RLC is the most potent PCNA-unloader among clamp-loader complexes.”

Page 20, line 14-16

Inserted following sentence.

“Although RFC could reduce the amount of DNA-loaded PCNA, its PCNA unloading activity is significantly lower than that of ATAD5-RLC (Supplementary Fig. 1k-n).”

Page 21, line 14-18

Inserted following sentence.

“Although RFC could unload PCNA, its PCNA unloading activity is significantly lower than that of ATAD5-RLC (Supplementary Fig. 1k-n). Different from ATAD5-RLC, previous reports showed that RFC-mediated PCNA unloading requires ATP hydrolysis¹⁰. RFC-mediated PCNA unloading was affected by DNA structure, while ATAD5-RLC could unload PCNA regardless of DNA structure (Supplementary Fig. 1h-j, m).”

Page 22, line 2-3

Inserted following sentence.

“It was reported that RFC also could unload mono-ubiquitinated PCNA⁴².”

Page 23, line 10-12

Inserted following sentences.

“It was reported that RFC-mediated PCNA unloading is also inhibited by polymerase δ ²⁰. Therefore, competition between clamp-loader complexes and PCNA-interacting proteins might serve as a general mechanism for PCNA cycling.”

Minor point 1

The authors claim that PCNA unloading requires the ATAD5-RLC complex, not just ATAD5 alone. This conclusion cannot be reached without an experiment where PCNA unloading is monitored in the presence of only the wild-type ATAD5 subunit.

As reviewer suggested, we purified ATAD5 (Δ N692) without RFC2-5 and compared its PCNA-unloading activity with that of ATAD5 (Δ N692)-RLC. We confirmed that ATAD5 (Δ N692) alone did not unload PCNA. This result strengthens our argument that ATAD5 should form RFC with RFC2-5 to unload PCNA. We revised our manuscript as below.

Page 10, line 23 ~ Page 11, line 3

“Therefore, PCNA unloading requires the ATAD5-RLC complex, not just ATAD5 alone.”

→ “To more clearly define functional contributions of each ATAD5 domain for PCNA unloading, we purified ATAD5 (Δ N692) alone and full-length ATAD5-RLC using the same system (Supplementary Fig. 2i) and examined their PCNA-unloading activity (Supplementary Fig. 2j). ATAD5 (Δ N692) alone did not show PCNA-unloading activity. PCNA unloading activity of full-length ATAD5-RLC and ATAD5 (Δ N692)-RLC were similar. Therefore, N-terminal domain of ATAD5 is not essential for PCNA unloading and PCNA unloading requires the ATAD5-RLC, not ATAD5 alone.”

Minor point 2

Functionality of the Cy3-labeled PCNA. The ability of PCNA to interact with other proteins, be loaded onto DNA, etc. is sensitive to amino acid mutations and modifications. To my knowledge, this is the first report where the native, exposed cysteines of PCNA were labeled with Cy3. It is customary to demonstrate that such labeling does not affect the ability of PCNA to function properly. Also, without correcting for the absorbance of Cy3 at 280 nm or determining the concentration of the protein via Bradford, the percent labeling is likely an underestimate.

In reply to Reviewer #1, we described our test of the loading and unloading activities of the modified PCNA. Regarding the labeling efficiency, following the suggestion of the reviewer,

we corrected it by measuring the protein concentration by Bradford assay and edited the text accordingly.

Page 40, line 9-12

“Labeling efficiency was measured by the ratio of Cy3 absorption at 550 nm ($150,000 \text{ M}^{-1}\text{cm}^{-1}$) and protein absorption at 280 nm ($15,930 \text{ M}^{-1}\text{cm}^{-1}$) to be 87% per target cysteine residue.”

→ “Labeling efficiency was calculated from Cy3 concentration measured by absorption at 550 nm ($150,000 \text{ M}^{-1}\text{cm}^{-1}$) and PCNA concentration measured by Bradford assay to show that each PCNA monomer contained 1.2 Cy3 molecules on average.”

Reviewer #1:

Remarks to the Author:

The authors fully addressed my previous concerns. It would be very informative though if the explanation that authors provided for the three-acceptor FRET is included as a supplemental note. This will be important for understanding the experimental design and for similar experiments in the future.

The addition of gel quantification by the authors strengthens the manuscript. The data, however, need to be presented, not just as mean +/- st.dev., but as scattered dot plot showing all measurements overlaid with mean +/- st.dev.

Reviewer #2:

Remarks to the Author:

The authors addressed most of the comments properly and the revised manuscript is potentially acceptable to Nature Commun.

However, I still have several questions to be answered in this revised form.

Comment 1.

The authors added a new protein purification method by the combination of yeast and bacterial expression system for human RFC, ATAD5-RLC or ATAD5 deletion mutants (page 32, lines 13-23). First of all, the method is confusing. They prepared yeast cells co-overexpressing RFC 1 or ATAD5 together with RFC2-5 once, and RFC2-5 were again supplied separately from E. coli lysates overexpressing them. Then, the full complex was reconstituted by mixing them together at 4°C. Did the authors actually apply such unusual method? Is the second supplementation of RFC2-5 necessary? Reasonable explanations should be necessary.

I am also confusing the sources of RFC and ATAD5-RLC used throughout this manuscript, since an old preparation method by baculovirus and a new one co-existed in this revised form. Please specify the sources of RFC and ATAD5-RLC proteins in results, respectively.

page 32, line 16; huamn should be human.

Comment 2. Difference between ATAD5-RLC and ATAD5 (Δ N692)-RLC

The authors emphasized the similarity of ATAD5-RLC and ATAD5 (Δ N692)-RLC. Indeed, their unloading activities are the same. But, effects of ATP- γ -S (Fig. 3e and Suppl. Fig. 3f) and mono-Ub of PCNA (Fig. 5c and Suppl. Fig. 5c) on their PCNA unloading seem to be different. ATP- γ -S did not affect PCNA unloading by ATAD5-RLC significantly (Suppl. Fig. 3f), although it interfered with the efficient PCNA unloading by ATAD5 (Δ N692)-RLC (Fig. 3e). The same result was observed by Elg1-RLC (Fig. 6a). Similarly, mono-Ub PCNA was obviously unloaded less efficiently by ATAD5-RLC than ATAD5 (Δ N692)-RLC (Suppl. Fig. 5c). These points should be mentioned in the text and the authors should describe a possible role to regulate PCNA unloading by the N-terminal region of ATAD5.

Comment 3. Effect of Dna2 on PCNA unloading Page 18 line 21 – Page 19 line 5

Although the data with Dna2 was presented in Fig. 6f, the results was not cited in this paragraph or in Discussion. Compared with other Okazaki fragment processing proteins, Dna2 does not have obvious PCNA binding motifs and its obvious physical binding to PCNA has not been reported. This property should be addressed to discuss the result of Dna2.

Reviewer #3:

Remarks to the Author:

The authors have responded to all my criticisms, and in my opinion, this new version of the paper can be published.

Reviewer's comment

Reviewer #1 (Remarks to the Author):

The authors fully addressed my previous concerns. It would be very informative though if the explanation that authors provided for the three acceptor FRET is included as a supplemental note. This will be important for understanding the experimental design and for similar experiments in the future.

Following this suggestion, we revised the smFRET part in Result section ("ATAD5-RLC binding to PCNA triggers PCNA release from DNA ") to explain the evidences that our multi-donor labeling scheme indeed allows to observe the average distance between PCNA ring and DNA substrate.

Page 12, line 12-15

"Real-time observation of PCNA loading revealed FRET dynamics followed by stepwise photobleaching of Cy3 dyes (Supplementary Fig. 4h). smFRET traces showing the fluorescence level of six Cy3 dyes were selected and the time range prior to the photobleaching of Cy3 dyes was used for further analysis."

Page 13, line 1-5

"FRET population maps built from PCNA loading traces with fewer Cy3 dyes exhibited broader FRET distribution but the major FRET levels were the same as those from traces with six Cy3 dyes, confirming that the FRET signal from our multi-donor labeling scheme can be interpreted as usual with single donor-acceptor pair (Supplementary Fig. 4i)."

The addition of gel quantification by the authors strengthens the manuscript. The data, however, need to be presented, not just as mean +/- st.dev., but as scattered dot plot showing all measurements overlaid with mean +/- st.dev.

As reviewer suggested, we overlaid data points as scattered dot blot in all bar charts presented in our manuscript.

Reviewer #2 (Remarks to the Author):

The authors addressed most of the comments properly and the revised manuscript is potentially acceptable to Nature Commun. However, I still have several questions to be answered in this revised form.

Comment 1.

The authors added a new protein purification method by the combination of yeast and bacterial expression system for human RFC, ATAD5-RLC or ATAD5 deletion mutants (page 32, lines 13-23). First of all, the method is confusing. They prepared yeast cells co-

overexpressing RFC1 or ATAD5 together with RFC2-5 once, and RFC2-5 were again supplied separately from E. coli lysates overexpressing them. Then, the full complex was reconstituted by mixing them together at 4°C. Did the authors actually apply such unusual method? Is the second supplementation of RFC2-5 necessary? Reasonable explanations should be necessary.

As reviewer pointed out, we found that supplement of bacterially over-expressed human RFC2-5 is necessary for the purification of ATAD5 or RFC1 in a complex from with RFC2-5. To avoid confusion, we added a sentence in Method section as follows.

Page 22, line 18-20

“Because co-overexpression of RFC2-5 was not sufficient to obtain RFC or RLC, RFC2-5 were supplemented using the bacterial expression system.”

I am also confusing the sources of RFC and ATAD5-RLC used throughout this manuscript, since an old preparation method by baculovirus and a new one co-existed in this revised form. Please specify the sources of RFC and ATAD5-RLC proteins in results, respectively.

We agree that different sources of protein purification might cause confusion. We described the sources of proteins throughout the result section.

Page 5, line 15-17

“To assess PCNA unloading activities of RFC and RLCs, we set up in vitro PCNA loading and unloading reactions with RFC and RLCs which are purified using Baculovirus system (Supplementary Fig. 1a).”

Page 6, line 8-9

“Based on the above results, we purified the N-terminal-deleted ATAD5 (ATAD5 (Δ N692)) in complex with RFC2-5 using the Baculovirus expression system (Fig. 1d).”

Page 6, line 23-24

“We prepared RFC containing full-length RFC1 using the combination of yeast and bacterial expression system (Supplementary Fig. 1a).”

Page 7, line 17-19

“Because previous studies reported PCNA-unloading activity of RFC¹⁸⁻²¹, we carefully compared PCNA-unloading activity of ATAD5 (Δ N692)-RLC and RFC1 (Δ N554)-RFC purified from the Baculovirus expression system (Supplementary Fig. 1k-n).”

Page 9, line 5-6

“Next, we purified an RLC-formation-defective mutant (693-1719) of ATAD5 using the Baculovirus expression system (Supplementary Fig. 2f).”

Page 9, line 6-11

“Because ATAD5 (693-1719) was less stable in insect cells, we also purified ATAD5 (693-1719) and ATAD5 (Δ N692)-RLC using the combination of yeast and bacterial expression system (Supplementary Fig. 2g). ATAD5 (693-1719) bound to a lesser amount of RFC2-5 compared to ATAD5 (Δ N692) and was defective in PCNA unloading in vitro (Fig. 2g (purified from Baculovirus system) and Supplementary Fig. 2h (purified from yeast-bacterial system)).”

Page 9, line 11-14

“Next, we purified ATAD5 (Δ N692) alone and full-length ATAD5-RLC using the yeast-bacterial system (Supplementary Fig. 2i) and compared their PCNA-unloading activity with ATAD5 (Δ N692)-RLC purified from the same system (Supplementary Fig.2).”

Page 10, line 16-17

“We purified ATPase-motif mutants (KA, NQA, and EK) with RFC2-5 using Baculovirus expression system (Supplementary Fig. 3b).”

Page 11, line 2-4

“We performed ATP- γ -S experiments with ATAD5 (Δ N692)-RLC and RFC1 (Δ N554)-RFC purified from the Baculovirus expression system.”

Page 11, line 8-11

“ATP- γ -S also interfered with efficient PCNA unloading by ATAD5-RLC (Fig. 3e (ATAD5 (Δ N692)-RLC purified from Baculovirus system) and Supplementary Fig. 3f (full-length ATAD5-RLC purified from the yeast-bacterial system))”

Page 11, line 16-18

“To mechanistically analyze PCNA loading and unloading by RFC and ATAD5-RLC, we performed smFRET experiments using ATAD5 (Δ N692)-RLC and RFC1 (Δ N554)-RFC prepared from Baculovirus expression system (Fig. 4a).”

Page 14, line 17-19

“We performed Ub-PCNA unloading assay with ATAD5 (Δ N692)-RLC purified from the Baculovirus expression system or full length ATAD5-RLC purified from the yeast-bacterial system.”

page 32, line 16; huamn should be human.

We corrected this typographical error.

Comment 2.

Difference between ATAD5-RLC and ATAD5 (Δ N692)-RLC The authors emphasized the similarity of ATAD5-RLC and ATAD5 (Δ N692)-RLC. Indeed, their unloading activities are the same. But, effects of ATP- γ -S (Fig. 3e and Suppl. Fig. 3f) and mono-Ub of PCNA (Fig. 5c and Suppl. Fig. 5c) on their PCNA unloading seem to be different. ATP- γ -S did not affect PCNA unloading by ATAD5-RLC significantly (Suppl. Fig. 3f), although it interfered with the efficient PCNA unloading by ATAD5 (Δ N692)-RLC (Fig. 3e). The same result was observed by Elg1-RLC (Fig. 6a). Similarly, mono-Ub PCNA was obviously unloaded less efficiently by ATAD5-RLC than ATAD5 (Δ N692)-RLC (Suppl. Fig. 5c). These points should be mentioned in the text and the authors should describe a possible role to regulate PCNA unloading by the N-terminal region of ATAD5.

We thank reviewer for the suggestion. We agree with the reviewers points that N-terminal domain of ATAD5 might regulate PCNA-unloading function. We included those points in Discussion section as follows.

Page 19, line 8-14

“Although the difference was not significant, full-length ATAD5-RLC unloaded mono-Ub-PCNA slightly less efficiently compared to unmodified PCNA (Supplementary Fig. 5c). In the

case of ATAD5 (Δ N692)-RLC, PCNA mono-ubiquitination did not affect PCNA-unloading efficiency (Fig. 5c). ATP- γ -S less affect PCNA-unloading activity of full-length ATAD5-RLC compared to that of ATAD5 (Δ N692)-RLC (Fig. 3e and Supplementary Fig. 3f). Therefore, it is possible that N-terminal domain of ATAD5-RLC has a role in the regulation of PCNA-unloading process.”

Comment 3.

Effect of Dna2 on PCNA unloading Page 18 line 21 – Page 19 line 5

Although the data with Dna2 was presented in Fig. 6f, the results was not cited in this paragraph or in Discussion. Compared with other Okazaki fragment processing proteins, Dna2 does not have obvious PCNA binding motifs and its obvious physical binding to PCNA has not been reported. This property should be addressed to discuss the result of Dna2.

We thank reviewer for this comment. We cite the result of Dna2 experiment and included discussion about the result as follows in Results section.

Page 17, line 2-3

“Dna2, which does not physically interact with PCNA, did not affect PCNA-unloading activity of Elg1-RLC.”

Reviewer #3 (Remarks to the Author):

The authors have responded to all my criticisms, and in my opinion, this new version of the paper can be published.

We thank reviewer.